



# A 30m resolution annual cropland extent dataset of Africa in recent decades of the 21st century

Zihang Lou[1], Dailiang Peng[2,3,*], Zhou Shi[1,*], Hongyan Wang[4], Yaqiong Zhang[5], Xue Yan[6,7], Zhongxing Chen[1], Su Ye[1], Le Yu[8], Jinkang Hu[2,3,10], Yulong Lv[2,3,10], Hao Peng[9,10], Yizhou Zhang[2,3,10], Bing Zhang[2,3,*]

[1]Institute of Applied Remote Sensing and Information Technology, College of Environmental and Resource Sciences, Zhejiang University, 866 Yuhangtang Road, Hangzhou 310058, China
[2]Key Laboratory of Digital Earth Science, Aerospace Information Research Institute, Chinese Academy of Sciences, Beijing 100094, China
[3]International Research Center of Big Data for Sustainable Development Goals, Beijing 100094, China
[4]Land Satellite Remote Sensing Application Center, Ministry of Natural Resources of P.R.China, Beijing 100048, China
[5]Satellite Application Center for Ecology and Environment, Ministry of Ecology and Environment, Beijing 100094, China
[6]Key Laboratory of Aquatic Botany and Watershed Ecology, Wuhan Botanical Garden, Chinese Academy of Sciences, Wuhan, 430074, China;
[7]Botany Department, Jomo Kenyatta University of Agriculture and Technology, P. O Box 62000, Nairobi, 00200, Kenya
[8]Department of Earth System Science, Tsinghua University, Beijing 100084, China
[9]State Key Laboratory of Desert and Oasis Ecology, Xinjiang Institute of Ecology and Geography, Chinese Academy of Sciences, Urumqi, Xinjiang 830011, China
[10]University of Chinese Academy of Sciences, Beijing 100094, China

*Correspondence to*: Dailiang Peng (pengdl@aircas.ac.cn), Zhou Shi (shizhou@zju.edu.cn) and Bing Zhang (zhangbing@aircas.ac.cn)

**Abstract.** Accurate cropland mapping is essential for understanding agricultural dynamics in Africa, a critical global issue with significant implications for the Sustainable Development Goals (e.g., Zero Hunger). Large-scale cropland mapping encounters several challenges, including the varying landscape characteristics of cropland across different regions, extended cultivation periods, and the limited availability of reference data. The study developed a 30-meter resolution African annual cropland distribution (namely AFCD) dataset for Africa spanning the years 2000 to 2022. To extract this large-scale cropland distribution data, we employed random forest classification and Continuous Change Detection algorithms on the Google Earth Engine platform. Robust training samples were generated, and a locally adaptive model was applied for cropland extraction. The final output consists of annual binary Crop/Non-Crop maps from 2000 to 2022. Independent validation samples from numerous third-party sources confirm that the map's accuracy is 0.86±0.01. A comparison of the cropland area estimates from AFCD with those of the FAO for Africa yielded an R-squared value of 0.86. According to our estimates, Africa's cropland expanded from 194.35 Mha in 2000 to 210.92 Mha by 2022, marking a net increase of 8.53%. Prior to 2005, changes in Africa's cropland area were gradual, but after 2006, there was a marked acceleration in cropland expansion. Despite this continued growth, Africa also experienced significant cropland abandonment. By 2018, abandoned cropland accounted for 11.52% of the total active cropland area. AFCD also avoided the misclassification of buildings, roads, and trees surrounding cropland, a common issue in LGRIP products. The study further highlights the unique advantage of AFCD in providing a



dynamic annual cropland dataset at 30-meter resolution for Africa. This dataset is a crucial resource for understanding the spatial-temporal dynamics of cropland and can support policies on food security and sustainable land management. The cropland dataset is available at https://doi.org/10.5281/zenodo.14920706 (Lou et al., 2025).

## 1 Introduction


According to the Food and Agriculture Organization (FAO), croplands play is of critical importance to global food sustainable development, and poverty alleviation. The FAO's State of Food Security and Nutrition in the World (SOFI) reported highlights that monitoring and managing agricultural lands was crucial for achieving the UN's Sustainable Development Goals, particularly in addressing food security challenges and balancing agricultural production with ecosystem services (FAO et al.,

2024). Agricultural land directly or indirectly contributes to 90% of global food calories (Cassidy et al., 2013). Beyond supplying food, farmland plays a crucial role as a provider of various ecosystem services(Pereira et al., 2018; Stephens et al., 2018), not only mitigating climate change through carbon sequestration and water and soil regulation (Lana-Renault et al., 2020), but also influencing biodiversity (Traba and Morales, 2019). Emerging agricultural paradigms prioritize climate-smart cultivation practices(Xiao et al., 2024), rational land-use policymaking (Duan et al., 2021), and productivity enhancement for

smallholder farmers. The expansion and abandonment of croplands have long been focal points of research, as the extent, distribution, and characteristics of cropland often influence a region's agricultural development pathways, food security, and poverty alleviation efforts (Jayne et al., 2014).

The SOFI report revealed that 20.4% of Africa's population faces hunger, with one in five people undernourished, severely hindering progress toward UN Sustainable Development Goals (SDGs) such as "No Poverty and Zero Hunger" and sustainable

agricultural-ecosystem balance (FAO et al., 2024; Ibrahim et al., 2023). While Africa accounted for 34% of global cropland expansion since 2000, driven by its perceived land abundance (Searchinger et al., 2015; Schneider et al., 2024), this growth had largely involved converting natural vegetation, causing significant deforestation and habitat destruction (Crawford et al., 2024; Kehoe et al., 2017). However, the economic and ecological trade-offs of such expansion remain understudied, necessitating nuanced land-use assessments (Chamberlin et al., 2014). Sub-Saharan Africa's food systems, dominated by

small-scale farms (<1 hectare), grapple with stagnant yields and rising imports (Fader et al., 2013; Giller et al., 2021) , while larger farms (>1 hectare) demonstrate higher productivity and food security through land consolidation (Nilsson, 2019). Yet unchecked agricultural intensification risks biodiversity and ecosystem stability (Adolph et al., 2023; Mano et al., 2020) , highlighting the urgency to monitor cropland dynamics. Detailed annual mapping of land-use changes is critical for guiding sustainable policies on food security, resource management, and environmental protection (Debonne et al., 2021; Waldner et

al., 2015), balancing Africa's agricultural needs with ecological preservation.

The rapid development of remote sensing technology has enabled accurate cropland mapping through synergistic use of multi-source satellite data. Moderate Resolution Imaging Spectroradiometer (MODIS) imagery offers unique advantages for large-scale cropland extent monitoring (Xiong et al., 2017a; Zhang et al., 2015, 2022), while higher-resolution sensors aboard



Landsat and Sentinel-2 satellites provide precise boundary delineation capabilities. Global and regional land use/land cover
(LULC) products, such as MCD12Q1 (Friedl and Sulla-Menashe, 2022), CCI Land cover (2017), GLC_FCS30D (Zhang et al., 2024b), and WorldCover, offer multi-scale data support for studying cropland distribution in Africa across a range of spatial resolutions (10-500 meters). However, their accuracy is generally limited, and they exhibit significant spatial inconsistencies (Cui et al., 2024; Song et al., 2022). These discrepancies arise from varying land cover definitions, temporal gaps due to cloud cover interference in remote sensing data, and the highly fragmented nature of African cropland landscapes
(Xiong et al., 2017c). While specialized cropland products, such as the Landsat Global Cropland Extent (Potapov et al., 2022), LGRIP (Teluguntla et al., 2023), GCEP (Xiong et al., 2017b), and Digital Earth Africa (Burton et al., 2022), offer high spatial resolution (ranging from 10 to 30 meters), allowing for detailed landscape characterization. However, their temporal coverage is often limited to single years or sparse intervals, which significantly restricts their ability to track rapid interannual changes in African agricultural systems (Kerner et al., 2024).

Key processes, such as cropland expansion into natural ecosystems and cyclical abandonment patterns, remain poorly understood and inadequately quantified due to these temporal gaps. Current LULC products are mainly based on single or multi-year remote sensing data, combined with expert-driven classification systems or machine learning algorithms such as random forests. However, challenges such as the scarcity of ground validation samples and significant surface heterogeneity in Africa hinder the improvement of classification accuracy. Notably, the successful application of sample generation methods
in crop recognition (Hu et al., 2024; Zhang et al., 2024a) and coastal zone monitoring (Zuo et al., 2025) demonstrates the potential of overcoming data bottlenecks through innovative sample construction techniques. Therefore, there is an urgent need to develop consensus label generation methods that integrate multiple LULC products (Kerner et al., 2024; Tubiello et al., 2023a). By leveraging the strengths of existing cropland datasets, this would enable the creation of high temporal resolution dynamic cropland maps for Africa. Furthermore, establishing a unified cropland classification standard system could
effectively reduce spatial inconsistencies across existing products, providing more accurate spatial information to support sustainable agricultural development and food security decision-making in Africa.

In summary, in recent decades, a lot of global or regional cropland mapping and monitoring has been made significant progress. However, a 30 m annual cropland extent time-series product derived from change-detection algorithms are still lacking. The aims of this study are (1) to construct locally optimal consensus Crop/Non-crop labels using LULC/cropland extent products
from 2000 to 2022, which will serve as training samples; (2)utilizing the continuous change-detection algorithm and the full time series of Landsat observations to generate the first African 30m annual cropland extent product that covers the period from 2000 to 2022, known as AFCD; (2) to quantitatively analyze the performance of AFCD product using multisource validation datasets.



## 2 Datasets

### 2.1 Previous LUCC & Cropland Datasets

To investigate the spatial distribution of Africa within the study's temporal scope, we amassed a comprehensive dataset from Google Earth Engine (GEE) and its community platform encompassing global or Africa-specific land use/cover change (LUCC) datasets as well as cropland distribution, with a spatial resolution finer than 500 meters, spanning the period from 2000 to 2022. However, definitions of cropland and other land use/cover types exhibit variations across different datasets. In this context,

both GLC_FCS30D and ESA CCI employ the FAO's Land Cover Classification System (LCCS) (ESA, 2017; Zhang et al., 2024b). The LCCS has been utilized to define diverse land cover types, demonstrating both flexibility and effectiveness across various geographical contexts (Ahlqvist, 2008) while simultaneously promoting interoperability for land-cover data and facilitating the scrutiny of classification processes (Herold et al., 2006). ESRI LUCC、ESA WorldCover and Dynamic World all employ a multi-tiered land cover classification system, defining cropland as areas planted or sown by humans with cereals,

grasses, and other crops that can be harvested within a year, excluding perennial woody crops (Karra et al., 2021; Van De Kerchove et al., 2021; Brown et al., 2022). Specialized cropland products define cropland as areas used for growing annual or perennial crops (Burton et al., 2022; Potapov et al., 2022), including forage and biofuel crops, under both rainfed and irrigated systems (Teluguntla et al., 2023), and also consider land permanently used for plantations like orchards and vineyards as cropland (Thenkabail et al., 2021).

**Table 1 Description of map data products utilized in this study.**

| Dataset | Year(s) | Res. (m/px) | Coverage | Reference |
|---|---|---|---|---|
| GLC_FCS30D | 1985; 1990; 1995; 2000-2022 | 30m | Global | (Zhang et al., 2024b) |
| ESRI 10m Annual Land Cover | 2017-2023 | 10m | Global | (Karra et al., 2021) |
| ESA WorldCover | 2020; 2021 | 10m | Global | (Zanaga et al., 2021, 2022) |
| Dynamic World | 2015-now | 10m | Global | (Brown et al., 2022) |
| ESA CCI | 1992-2020 | 300m | Global | (ESA, 2017) |
| Digital Earth Africa Cropland Extent | 2019 | 10m | Continent | (Burton et al., 2022) |
| GFSAD global cropland maps | 2015 | 30m | Global | (Thenkabail et al., 2021) |
| GLAD Global Cropland Maps | 2003, 2007, 2011, 2015, 2019 | 30m | Global | (Potapov et al., 2022) |
| Africa Cropland Mask | 2016 | 30m | Continent | (Nabil et al., 2022) |

Earth System Science Data Discussions Open Access

## 2.2 Continuous Landsat imagery from 2000 to 2022

All available surface reflectance (SR) from Landsat imagery (Level 2, Collection 2, Tier 1) spanning 2000 to 2022, including Landsat 5, 7, 8, and 9, stored in the GEE computational platform, were collected to monitor the spatiotemporal dynamics of
cropland extent in Africa. To minimize the spectral discrepancies among different Landsat sensors, calibration coefficients were applied to recalibrate the surface reflectance data from TM and ETM+ to align with the comparable standards of OLI (Roy et al., 2016). Subsequently, the Landsat Ecosystem Disturbance Adaptive Processing System (LEDAPS) as well as the Land Surface Reflectance Code (LaSRC) algorithms were employed to perform atmospheric correction on Landsat imagery and construct a high-quality, continuous Landsat temporal dataset (Vermote and Saleous, 2007; Vermote and Kotchenova,
125     2008).

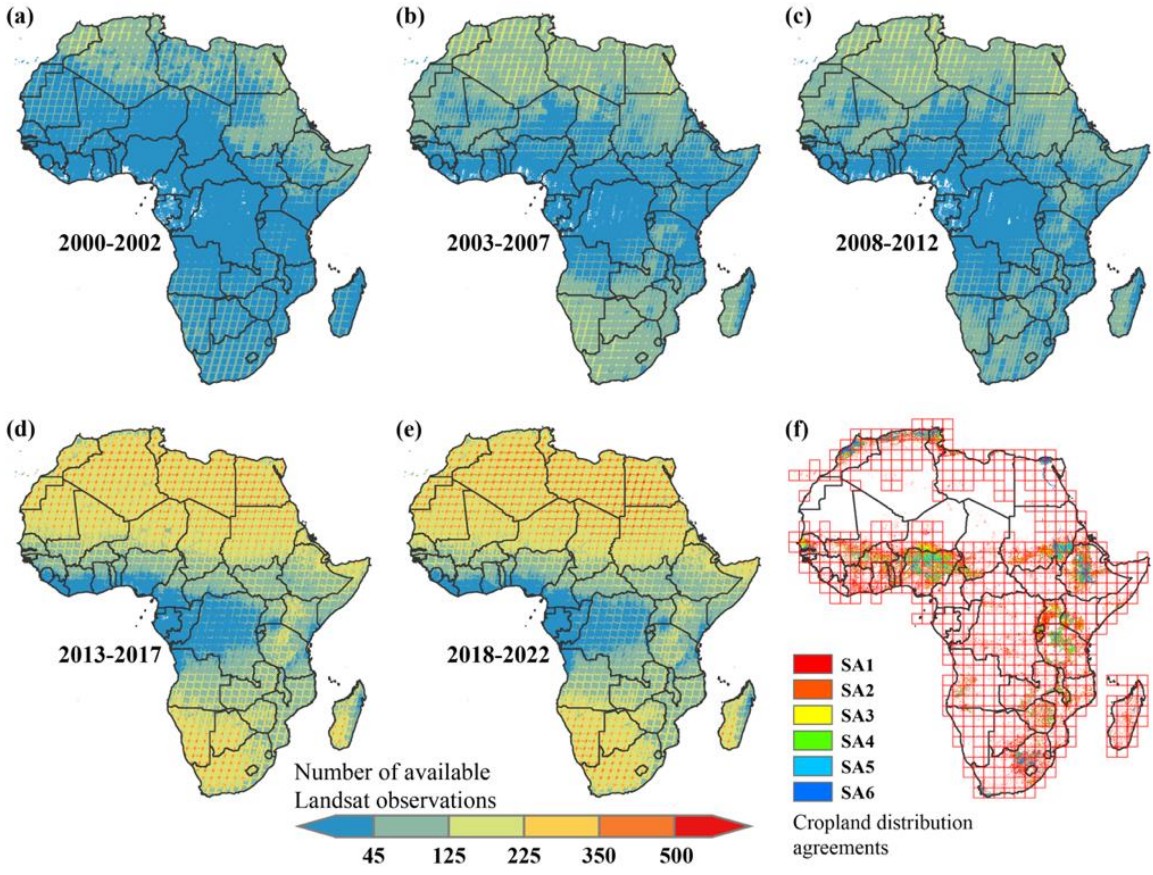

**Figure 1** Temporal and Spatial Distribution of Available Landsat Imagery Data Over Africa and (a-e) and Spatial Distribution of Cropland (f) (Tubiello et al., 2023a, b).

### 2.3 Validation datasets

To comprehensively analyze the accuracy metrics of the AFCD map, we collected two validation datasets: cropland samples across Africa in 2016 and 2019. Initially, we utilized 3,386 independent validation samples provided by Kerner et al. (2024), collected through field surveys and visual interpretation across eight countries in sub-Saharan Africa (shown as Fig. 2 (b)). Additionally, we utilized crowdsourced data provided by (Laso Bayas et al., 2017), collected via the Geo-Wiki platform, for validation (Fig. 2 (a)). The dataset selected cropland samples within the African region corresponding to the 300m x 300m

grid of PROBA-V imagery. Globally, 35,866 samples were set, with 7,313 located in Africa. Based on participant collection, samples were randomly selected for further validation by students and experts. Each sample recorded the proportion of cropland area within the 300m x 300m frame.

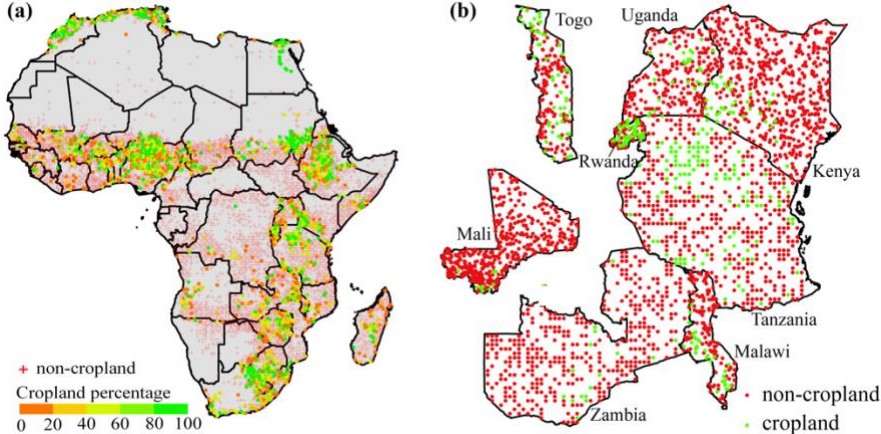

**Figure 2 Spatial distribution of validation points for regional and time-series.**


## 3. Methodology

As the framework shows in Fig. 2, we have proposed a landcover change detection-based approach that combines machine learning and continuous change detection algorithm for mapping annual cropland extent in Africa. Therefore, we defined the cropland as land used for annual and perennial herbaceous crops for human consumption, forage (including hay) and biofuel.

Perennial woody crops, permanent pastures, and crop rotations are excluded from the definition. The identification of active cropland relies on significant variations in vegetation signals within time-series remote sensing data over a 12-month period, indicating planting and harvesting activities. When periodic vegetation signals exhibit anomalies, we hypothesize that changes in cropping patterns or systems may have occurred.





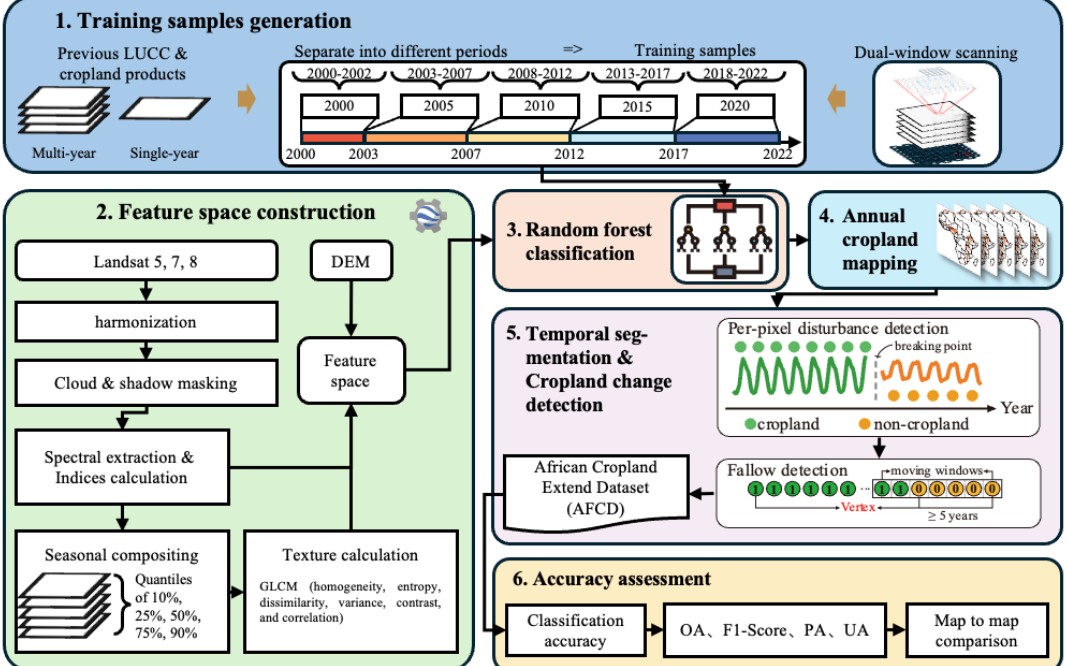

Figure 3 Flowchart of this study for mapping African annual cropland extent spanning the years 2000 to 2022.

## 3.1 Generate training datasets

In the process of generating training data, we employed a dual-window sliding window strategy for the creation of cropland and non-cropland training samples. Prior studies have revealed that the distribution of training data (proportional to area and equally distributed) as well as data balance significantly impact classification outcomes. Quantitative analyses have demonstrated that the proportional allocation method typically achieves higher overall accuracy compared to an equally distributed approach (Jin et al., 2014; Zhu et al., 2016). Specifically, our procedure for generating reference samples is as follows:

- All annual and multi-year products were categorized into five groups, centred around the years 2000, 2005, 2010, 2015, and 2020, each group encompassing the product from the respective centre year plus or minus two years.
- After remapping the classification systems of different products, the categories were binarized (excluding categories with significant discrepancies from other products). The data were then resampled to the WGS84 coordinate system with a spatial resolution of 0.0002875 degrees (approximately 30 meters at the equator). For multi-year products, areas where various land cover types had not changed were extracted; for single-year products, they were used directly.
- A dual-window methodology was applied to assess binary image classification, requiring complete category occupancy within a 150m×150m inner window for homogeneity, and at least 80% dominance in a 330m×330m outer window, ensuring spatial stability of the central pixel as a representative sample point.




- An evaluation was performed to ascertain the recognition consistency of the central pixel's classification across various products.
- A hexagonal grid dataset with a resolution of 0.01 degrees was employed to randomly filter the generated excess sample
points. Within each grid cell, a maximum of 10 samples of the same category with the highest recognition level were retained.

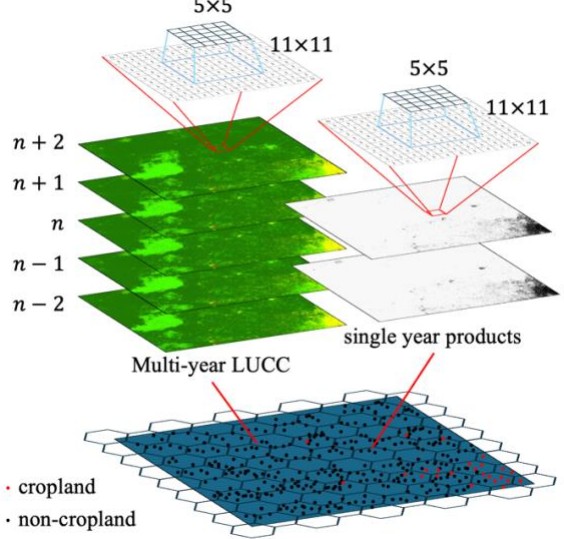

**Figure 4** Schematic Diagram of Training Sample Dataset Generation Using a Dual-Window Approach.

Through the procedures, a total of five sets of sample datasets were generated, each corresponding to the classification sample
data for a central year and the two preceding and following years.

### 3.2 Cropland Mapping with classification methods

### 3.3.1 Original cropland mapping

The temporal scope for cropland extraction was extended from 2000 to 2022. Utilizing multi-temporal Landsat Surface Reflectance (SR) imagery is a primary method for preprocessing in contemporary land use classification. This approach
mitigates the limitations of single-scene imagery by facilitating the extraction of seasonal changes on the Earth's surface. Consequently, two methods for temporal data synthesis have been developed: seasonal composites (e.g., 16-day, monthly, and seasonal composites) and metrics composites. The metrics composite method, introduced by Hansen et al.(2013), captures phenological and land cover changes without requiring assumptions or prior knowledge about seasonal timing, making it globally applicable without location-specific modifications.
In this study, five spectral bands from Landsat, excluding the blue band, were used, along with four spectral indices: Enhanced Vegetation Index (EVI), Soil-Adjusted Vegetation Index (SAVI), Normalized Difference Water Index (NDWI), and Normalized Burn Ratio (NBR). From these nine spectral features, five percentiles (10th, 25th, 50th, 75th, and 90th) were





calculated, resulting in a total of 45 spectral features. Additionally, topographic parameters (elevation, slope, and aspect) were derived from the global 30m Digital Elevation Model (DEM) provided by ASTER GDEM. Texture features (variance,

homogeneity, contrast, dissimilarity, entropy, and correlation) were generated from the gray-level co-occurrence matrix based on the Near Infrared (NIR) band, yielding a total of 54 feature parameters for subsequent classification tasks.

For classifier selection, we utilized the ee.Classifier.smileRandomForest() method available on the GEE cloud platform. This algorithm determines pixel class membership by adjusting two key parameters: the number of decision trees (Ntree) and the number of predictor variables (Mtry) at each node split, using a combination of training samples and multivariate features.

Previous studies have shown that the classification accuracy of this algorithm is not significantly affected by the specific parameter values. Therefore, Ntree was set to the default value of 500, and Mtry was set to the square root of the total number of input features. A separate classifier was trained for each 2° grid (Fig. 1 f) cell using the Random Forest method, with training samples drawn from the four neighboring grid cells surrounding the target grid unit.

### 3.3.2 Change detection by CCD algorithm and updating changes

Changes in cropland are generally more complex than other forms of land use changes, such as deforestation and urban expansion. These changes can be categorized as follows: (1) Acquisition of new agricultural land, achieved through clearing forests or converting savannah regions into arable land; (2) Short-term fallowing, which involves temporarily resting the land to restore soil fertility; (3) Abandonment, where agricultural land is left unused due to various factors; and (4) Urban replacement, where agricultural fields are progressively replaced by urban areas as a result of socio-economic development.

In this study, the CCD algorithm (Zhu and Woodcock, 2014) was applied to identify changes in cropland areas. This algorithm employs Fourier transform techniques to model time-series observations using trend terms (to estimate trend changes) and harmonic terms (to describe periodic changes), thereby facilitating precise detection of changes in cropland regions. Specifically, the algorithm first decomposes the time-series data into trend and periodic components using the Fourier transform, as shown in Eq. (1):

$$\hat{\rho}(i,t)\& = a_{0,i} + c_{1,i} \times t + \sum_{k=1}^{n} \left( a_{k,i} \times \cos\left(\frac{2k\pi}{T}t\right) + b_{k,i} \times \sin\left(\frac{2k\pi}{T}t\right)\right), \tag{1}$$

Where $\hat{\rho}(i,t)$ represents the estimated value at Julia day $t$ and band $i$. The parameter $a_{0,i}$ is the intercept term at band $i$, reflecting the average level of the time series, while $c_{1,i}$ is the linear trend coefficient, indicating the linear change over time at that location. The summation $\sum_{k=1}^{n}$ includes the harmonic components, where $n$ is the number of harmonics. The coefficients $a_{k,i}$ and $b_{k,i}$ are the cosine and sine coefficients for the $k$-th harmonic, describing the amplitude and phase of

periodic changes. The functions $\cos\left(\frac{2k\pi}{T}t\right)$ and $\sin\left(\frac{2k\pi}{T}t\right)$ are the cosine and sine functions for the $k$-th harmonic, with T being the period length (default is 365.25), used to capture periodic variations (Xian et al., 2022).

In this study, we utilized the JavaScript API provided by the GEE platform, specifically the ee.Algorithms.TemporalSegmentation.Ccdc() function, to detect land use changes. This algorithm identifies change points in land use by analysing time-series data. In this study, we applied the CCD algorithm alongside continuous Landsat imagery





data to determine the stable time intervals of agricultural land pixels and identify specific temporal points of land use changes. This approach also enabled the identification and correction of misclassifications occurring during these stable periods by majority voting.

### 3.4 Accuracy assessment

The validation of the African cropland product involved three complementary approaches. First, we conducted cross-
comparison with national cropland statistics from the FAO reports to assess area estimation accuracy. Second, we performed sample-based validation using multiple independent datasets. Third, we implemented spatial consistency analysis by comparing our product with existing remote sensing-derived cropland maps. For the sample-based validation, we employed three distinct strategies: (1) We reserved 30% of consensus samples from model training for internal validation of classification accuracy; (2) Utilized the cropland/non-cropland samples from sub-Saharan Africa compiled by Kerner et al., (2024) for
independent accuracy assessment (as described in 2.3); and (3) Applied the global cropland sample dataset from Laso Bayas et al., (2017) for sample-based area estimation validation (as described in 2.3). The first two validation approaches employed four key accuracy metrics calculated through confusion matrix analysis, as shown in Eq. (2):

$$
\begin{aligned}
PA_k &= \frac{p_{kk}}{p_{k\cdot}} \\
UA_k &= \frac{p_{kk}}{p_{\cdot k}} \\
OA &= \frac{\sum_{k=1}^{m} p_{kk}}{N} \\
F1_k &= \frac{2 \cdot PA_k \cdot UA_k}{PA_k + UA_k} \\
Kappa &= \frac{N \sum_{k=1}^{m} p_{kk} - \sum_{k=1}^{m} p_{k\cdot} p_{\cdot k}}{N^2 - \sum_{k=1}^{m} p_{k\cdot} p_{\cdot k}}
\end{aligned}
\tag{2}
$$

Let $p_{ij}$ denote the number of samples belonging to class $i$ but classified as class $j$, with $p_{kk}$ representing correctly classified
cases. The marginal sums $p_{k\cdot} = \sum_j p_{kj}$ and $p_{\cdot k} = \sum_i p_{ik}$ correspond to row and column totals respectively.

For the Laso Bayas et al. (2017) dataset validation, we calculated mean absolute error (MAE) by comparing our mapped cropland area proportions against reference values within 300×300m sample units. The MAE computation is shown in Eq. (3):

$$
\text{MAE} = \frac{1}{n} \sum |A_m - A_r|
\tag{3}
$$

Where $A_m$ denotes the mapped cropland proportion, $A_r$ is the reference proportion, and n is the sample count.

In addition, we evaluated the similarity of AFCD with other multi-source remote sensing cropland extent maps through comparative analysis. First, five representative agricultural zones were selected as sample areas based on farmland intensification levels and spatial continuity characteristics, including: (1) contiguous irrigated agricultural zones, (2) dispersed traditional rain-fed agricultural areas, (3) mountainous terrace farming systems, (4) semi-intensive agro-pastoral transition areas, and (5) fragmented transitional farmland regions (see Figure 2). Within each sample area, 1,500 validation points were
systematically collected, with similarity matrices (Phalke et al., 2020) employed to compare our maps with other remote sensing-based land cover products. It should be specifically noted that the comparative datasets (e.g., GLAD, as described in



Section 2.1) exhibit significant discrepancies in farmland definitions, spatial resolution, mapping years, and classification methodologies. To mitigate the impacts of these disparities during comparison, spatial resampling of our map products was occasionally required, along with temporal alignment using outputs from identical reference years as comparator products.

**4 Results**

**4.1 Classification results**

The result of this study is developing a new annual cropland dynamic map of Africa at 30-m (Fig. 5). The visual evaluation of the current cropland product shows that cultivated areas are accurately represented across diverse agricultural landscapes throughout Africa (Fig. 5). As shown in the figure above, five regions were selected to assess the AFCD's recognition

capabilities, each representing different agricultural characteristics. The contiguous irrigated agricultural zones in Egypt, characterized by large, contiguous cropland due to intensive agriculture, were easily identified (Fig. 5b). Similarly, the dispersed traditional rain-fed agricultural areas in Senegal were accurately captured (Fig. 5b). The mountainous terrace farming systems in Rwanda were also well mapped (Fig. 5b), along with the semi-intensive agro-pastoral transition areas in Kenya (Fig. 5b). Lastly, the fragmented transitional farmland regions in Sudan were correctly identified, showcasing the AFCD's

capacity to detect various agricultural systems across different landscapes.



**Figure 5** Cropland Extent Map of Africa in 2022 at 30m Resolution by the Africa Cropland Dynamic map (AFCD) Project (a); Visual Interpretation of Cropland Extent for Selected African Countries (b), and interannual variations in cropland areas across Africa during the 21st (c). Yellow colour represents the cropland mask, and background is the natural colour composite high-resolution imagery available in © Google Earth Engine.

## 4.1 Independent accuracy assessment of AFCD map

We performed pixel-wise accuracy assessment of annual cropland map based on the validation sample set. The F1 score, OA, UA, PA, and Kappa coefficient of annual maps on average were $0.86 \pm 0.01$, $0.97 \pm 0.01$, $0.88 \pm 0.07$, $0.77 \pm 0.07$, and $0.93 \pm 0.01$, respectively. Spatially, mapping accuracy can vary across different tiles, likely due to the sparse distribution of cropland. The overall classification accuracy of the model exceeds 90% across most regions, except for central and southern Niger, where the accuracy is lower. In most areas, the F1 scores are above 0.80, and both omission and commission errors are less





than 0.2. For the sample data provided by Kerner for eight countries in sub-Saharan Africa, the sample collection year is 2019. Consequently, we selected cropland extraction results from this time frame for validation. The outcomes and comparisons with other products are presented in Table 1. We primarily utilized this sample to evaluate our product, while the accuracy metrics

for other products were sourced from the study by Kerner et al., (2024). Based on four evaluation metrics and results from eight target countries, we compared the performance of our product against other nations and the average level. Our cropland identification model performed exceptionally well in countries such as Kenya and Mali, achieving an accuracy of 0.96 and an F1 score of 0.92, indicating a strong balance between precision and recall. However, the model's performance was relatively weaker in Zambia, Togo, and Uganda, with accuracies of 0.82 and 0.78, respectively.

**Table 2 Performance metrics and associated standard errors for our results and other maps based Kerner's reference dataset. The highest value in each row is highlighted in bold blue and the second highest in bold black.**

| Country | Metric | Our Result | GFSAD | GLAD | Copernicus | GlobCover | ASAP | Mean |
|---|---|---|---|---|---|---|---|---|
| Kenya | Accuracy | **0.96 ± 0.00** | 0.92 ± 0.01 | **0.95 ± 0.01** | 0.90 ± 0.01 | 0.76 ± 0.01 | 0.92 ± 0.01 | 0.91 ± 0.05 |
| | F1 | **0.66 ± 0.02** | 0.63 ± 0.14 | **0.75 ± 0.19** | 0.50 ± 0.18 | 0.29 ± 0.11 | 0.62 ± 0.17 | 0.55 ± 0.13 |
| | Precision (UA) | **0.67 ± 0.03** | 0.47 ± 0.06 | **0.68 ± 0.07** | 0.38 ± 0.06 | 0.18 ± 0.03 | 0.48 ± 0.07 | 0.48 ± 0.16 |
| | Recall (PA) | 0.64 ± 0.02 | **0.95 ± 0.03** | 0.83 ± 0.05 | 0.73 ± 0.07 | 0.74 ± 0.07 | **0.85 ± 0.05** | 0.73 ± 0.15 |
| Malawi | Accuracy | 0.83 ± 0.00 | 0.77 ± 0.03 | **0.86 ± 0.03** | 0.84 ± 0.03 | 0.75 ± 0.03 | 0.77 ± 0.03 | 0.81 ± 0.04 |
| | F1 | 0.61 ± 0.01 | 0.63 ± 0.19 | **0.66 ± 0.24** | **0.67 ± 0.22** | 0.50 ± 0.23 | 0.63 ± 0.17 | 0.59 ± 0.14 |
| | Precision (UA) | 0.55 ± 0.01 | 0.50 ± 0.07 | **0.71 ± 0.09** | 0.61 ± 0.08 | 0.43 ± 0.08 | 0.48 ± 0.07 | 0.58 ± 0.09 |
| | Recall (PA) | 0.68 ± 0.03 | 0.85 ± 0.05 | 0.62 ± 0.07 | 0.74 ± 0.07 | 0.61 ± 0.08 | **0.92 ± 0.05** | 0.67 ± 0.23 |
| Mali | Accuracy | 0.94 ± 0.01 | 0.91 ± 0.01 | **0.96 ± 0.01** | 0.93 ± 0.01 | 0.80 ± 0.01 | **0.95 ± 0.01** | 0.92 ± 0.05 |
| | F1 | **0.23 ± 0.07** | 0.15 ± 0.20 | **0.33 ± 0.29** | 0.18 ± 0.23 | 0.07 ± 0.08 | 0.21 ± 0.38 | 0.21 ± 0.14 |
| | Precision (UA) | 0.16 ± 0.05 | 0.10 ± 0.05 | 0.23 ± 0.08 | 0.12 ± 0.06 | 0.04 ± 0.02 | 0.17 ± 0.11 | 0.14 ± 0.10 |
| | Recall (PA) | 0.42 ± 0.13 | 0.37 ± 0.15 | **0.61 ± 0.16** | 0.37 ± 0.15 | 0.42 ± 0.16 | 0.27 ± 0.15 | 0.44 ± 0.25 |
| Rwanda | Accuracy | **0.76 ± 0.00** | 0.74 ± 0.04 | **0.77 ± 0.04** | **0.77 ± 0.04** | 0.61 ± 0.05 | 0.71 ± 0.05 | 0.75 ± 0.07 |
| | F1 | 0.63 ± 0.01 | 0.73 ± 0.18 | 0.72 ± 0.18 | **0.75 ± 0.18** | 0.61 ± 0.18 | **0.74 ± 0.16** | 0.69 ± 0.11 |
| | Precision (UA) | **0.82 ± 0.02** | 0.67 ± 0.07 | **0.84 ± 0.07** | 0.73 ± 0.07 | 0.53 ± 0.07 | 0.64 ± 0.07 | 0.77 ± 0.14 |
| | Recall (PA) | 0.51 ± 0.01 | 0.79 ± 0.05 | 0.64 ± 0.05 | 0.78 ± 0.05 | 0.72 ± 0.05 | **0.88 ± 0.04** | 0.68 ± 0.17 |
| Tanzania | Accuracy | 0.83 ± 0.01 | **0.88 ± 0.01** | 0.86 ± 0.01 | 0.86 ± 0.01 | 0.70 ± 0.01 | 0.83 ± 0.01 | 0.81 ± 0.06 |
| | F1 | 0.72 ± 0.02 | **0.76 ± 0.06** | 0.69 ± 0.05 | 0.72 ± 0.06 | 0.51 ± 0.06 | 0.65 ± 0.05 | 0.61 ± 0.16 |
| | Precision (UA) | 0.89 ± 0.02 | 0.89 ± 0.02 | **0.95 ± 0.01** | 0.86 ± 0.02 | 0.60 ± 0.03 | 0.92 ± 0.02 | 0.87 ± 0.11 |
| | Recall (PA) | **0.61 ± 0.02** | **0.67 ± 0.02** | 0.54 ± 0.02 | **0.61 ± 0.02** | 0.45 ± 0.02 | 0.50 ± 0.02 | 0.49 ± 0.16 |
| Togo | Accuracy | 0.77 ± 0.01 | 0.77 ± 0.03 | **0.86 ± 0.03** | **0.78 ± 0.03** | 0.74 ± 0.03 | 0.69 ± 0.03 | 0.78 ± 0.06 |
| | F1 | 0.45 ± 0.03 | 0.64 ± 0.18 | **0.75 ± 0.15** | 0.63 ± 0.19 | 0.46 ± 0.20 | 0.49 ± 0.20 | 0.55 ± 0.20 |
| | Precision (UA) | 0.68 ± 0.02 | 0.60 ± 0.07 | **0.88 ± 0.06** | 0.64 ± 0.07 | 0.56 ± 0.09 | 0.48 ± 0.07 | 0.70 ± 0.14 |
| | Recall (PA) | 0.33 ± 0.03 | **0.68 ± 0.05** | 0.66 ± 0.05 | 0.62 ± 0.05 | 0.38 ± 0.05 | 0.51 ± 0.06 | 0.50 ± 0.22 |
| Uganda | Accuracy | **0.86 ± 0.01** | 0.79 ± 0.02 | **0.84 ± 0.02** | 0.77 ± 0.02 | 0.57 ± 0.02 | 0.70 ± 0.02 | 0.78 ± 0.10 |
| | F1 | 0.53 ± 0.02 | 0.48 ± 0.21 | **0.57 ± 0.23** | 0.40 ± 0.20 | 0.31 ± 0.12 | 0.38 ± 0.16 | 0.44 ± 0.08 |
| | Precision (UA) | 0.42 ± 0.02 | 0.35 ± 0.07 | 0.46 ± 0.08 | 0.29 ± 0.09 | 0.19 ± 0.04 | 0.25 ± 0.05 | 0.38 ± 0.15 |
| | Recall (PA) | 0.72 ± 0.01 | 0.73 ± 0.07 | 0.76 ± 0.07 | 0.67 ± 0.09 | **0.80 ± 0.07** | 0.79 ± 0.07 | 0.66 ± 0.18 |
| Zambia | Accuracy | **0.96 ± 0.00** | 0.94 ± 0.01 | **0.97 ± 0.01** | 0.94 ± 0.01 | 0.90 ± 0.01 | 0.91 ± 0.01 | 0.94 ± 0.03 |
| | F1 | 0.51 ± 0.06 | 0.60 ± 0.23 | **0.73 ± 0.27** | 0.58 ± 0.24 | 0.20 ± 0.23 | 0.45 ± 0.21 | 0.56 ± 0.15 |
| | Precision (UA) | 0.47 ± 0.05 | 0.46 ± 0.09 | **0.68 ± 0.10** | 0.45 ± 0.09 | 0.15 ± 0.06 | 0.32 ± 0.07 | 0.49 ± 0.19 |
| | Recall (PA) | 0.58 ± 0.08 | **0.85 ± 0.07** | 0.79 ± 0.08 | **0.80 ± 0.08** | 0.29 ± 0.10 | 0.78 ± 0.08 | 0.71 ± 0.17 |
| Mean | Accuracy | **0.86 ± 0.01** | 0.84 ± 0.01 | **0.88 ± 0.01** | 0.85 ± 0.01 | 0.73 ± 0.01 | 0.81 ± 0.01 | - |
| | F1 | 0.54 ± 0.03 | **0.58 ± 0.05** | **0.65 ± 0.08** | 0.55 ± 0.06 | 0.37 ± 0.07 | 0.52 ± 0.09 | - |
| | Precision (UA) | **0.58 ± 0.03** | 0.51 ± 0.02 | **0.68 ± 0.03** | 0.51 ± 0.02 | 0.34 ± 0.03 | 0.47 ± 0.03 | - |
| | Recall (PA) | 0.56 ± 0.04 | **0.74 ± 0.04** | 0.68 ± 0.04 | 0.67 ± 0.04 | 0.55 ± 0.04 | **0.69 ± 0.04** | - |

Secondly, we utilized the crowd sourced sample data provided by Laso Bayas et al. (2017) to extract the proportion of cropland area within a 300m × 300m radius around each point. We then calculated the MAE between the extraction results and visual

interpretation, the result indicated that 15.07 ± 1.70% for all samples, 10.78 ± 1.56% for control samples, and 31.45 ± 8.47% for expert samples.

### 4.2 Comparison of cropland areas

The net cropland area for Africa were estimated from 19238.65 Mha in 2000 to 21092.43 Mha in 2022. This estimate is in line with other remote sensing-based estimates, although major differences exist. This study conducted cross-validation between

national-scale cultivated land area estimations and statistical data from the FAO covering 50 African countries/regions (partial areas excluded due to data gaps), demonstrating strong concordance ($R^2$ = 0.83, Fig. 6a). It should be specifically noted that

our total farmland estimates exhibit a 10%–24% discrepancy range compared with official FAO statistics. This divergence may originate from two principal factors: (1) FAO's statistical framework heavily relies on self-reported national data, inherently constrained by inconsistent survey methodologies and update delays (Fritz et al., 2015); (2) our remote sensing interpretation protocol using growing-season NDVI threshold classification potentially underestimates intermittently fallowed croplands. Comparative analysis with independent remote sensing products (GLAD and IGRIP) revealed three critical findings: First, the 30-m resolution GLAD dataset showed optimal linear agreement with our estimates (intercept = $0.30 \times 10^3$ ha, $R^2$ = 0.94, Fig. 6b). Second, the 100-m resolution IGRIP dataset exhibited a larger intercept of $1.19 \times 10^3$ ha ($R^2$ = 0.86, Fig. 6c). Notably, the minimal area discrepancy between GLAD and our study (slope coefficient = $0.98 \pm 0.03$) primarily stems from mutual adoption of 30-m resolution detection criteria. However, the observed high consistency with IGRIP ($R^2$ = 0.86) mainly reflects mutual recognition of dominant farmland types, given IGRIP's emphasis on rainfed/irrigated classifications (unaddressed in our study) and Africa's overwhelming predominance of rainfed cultivation systems (FAO, 2020).

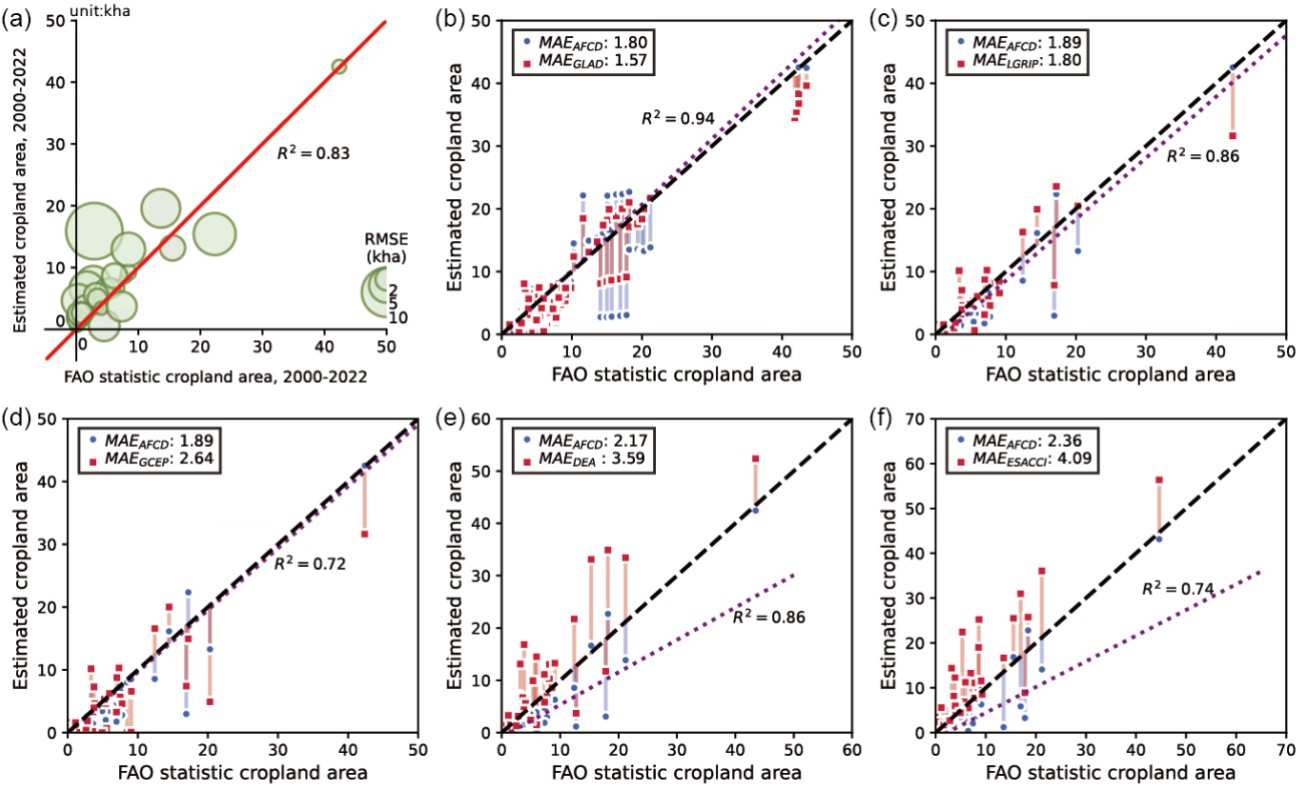

**Figure 6** Comparison of AFCD with FAO Statistical Cropland Area and Other LULC/Cropland Products (2000–2022)

### 4.3 Map to map comparison results

The comparison of cropland maps derived from this study with the cropland labels in GLAD2019 reveals a high level of agreement across the entire study area, with an overall similarity of 87.7%. Specifically, the producer's cropland similarity



was 85.2% (corresponding to an omission error of 14.8%), while the user's cropland similarity was slightly higher at 86.9% (resulting in a commission error of 13.1%). Among the selected regions, Nigeria exhibited the highest similarity at 94.6%, with an F1-Score of 0.963. Senegal followed closely with an overall similarity of 94.4% (F1-Score = 0.858), while Rwanda showed the lowest similarity at 76.7% (F1-Score = 0.766). Visual comparisons highlighted the successful identification of major irrigated agricultural zones (such as in Egypt) and rainfed cultivation areas (notably in Nigeria).

When compared with the 10m African cropland product from Digital Earth Africa (DEA) for 2019, the similarity in Egypt was particularly high, with an overall similarity of 90.1% (F1-Score = 0.913). In this case, the producer's cropland similarity was 86.5% (omission error = 13.5%), while the user's cropland similarity reached 96.5% (commission error = 3.5%). The primary cause of discrepancies is the use of Sentinel-2 10m-resolution imagery in the DEA product, which improves boundary delineation between cropland and adjacent land covers, reducing mixed-pixel effects. As a result, our AFCD product tends to show more omission errors compared to DEA, while the commission errors remain below 5%, highlighting its strong performance in detecting cropland areas.

A cross-comparison with the LGRIP30 global 30m irrigated/rainfed cropland dataset (2015 vintage) in Nigeria's predominantly rainfed systems showed an overall similarity of 89.7% (F1-Score = 0.932). The producer's cropland similarity was notably high at 95.8% (omission error = 4.2%), while the user's cropland similarity was 88.4% (commission error = 11.6%). The discrepancy is mainly due to occasional misclassifications in the LGRIP30 dataset, where buildings, roads, and neighboring woodlands are sometimes incorrectly labeled as cropland. Nevertheless, the spatial agreement between the AFCD and LGRIP30 datasets remains substantial, with a similarity of nearly 90%.

Earth System
Open Access · Science
Data · Discussions

**Figure 7** Visual comparison of cropland and non-cropland classification results from AFCD and Potapov et al. (glad) across multiple African countries, and background is the natural colour composite high-resolution imagery available in © Google Earth Engine.



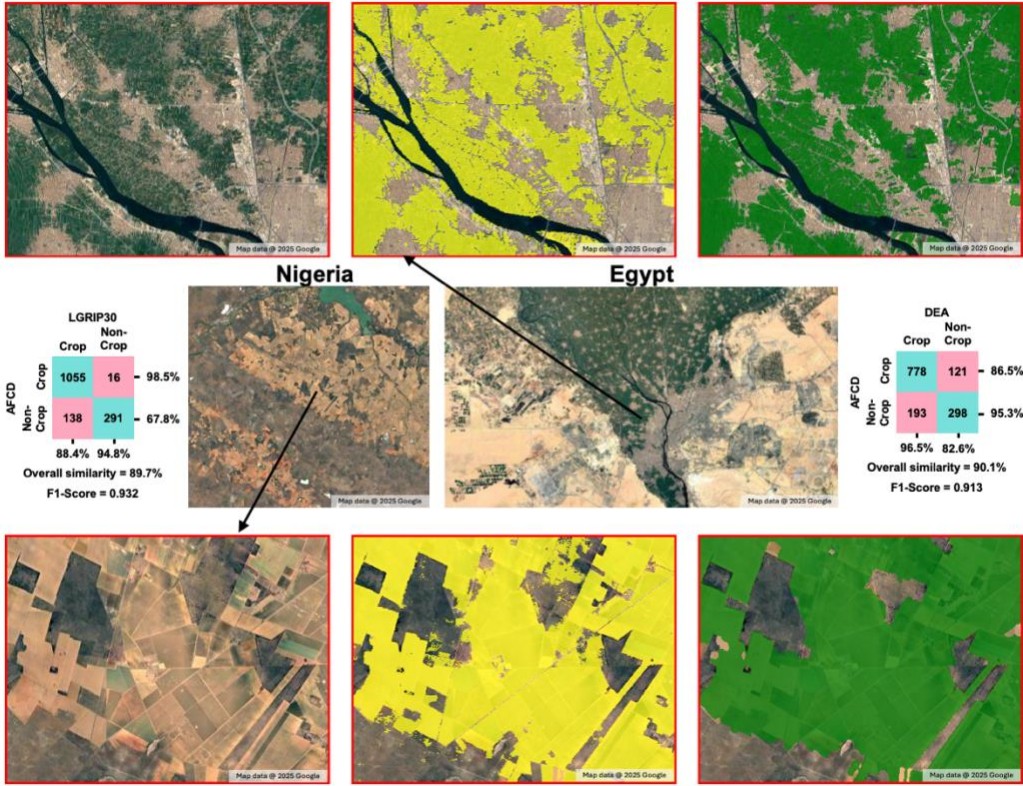

**Figure 8** Map to map comparison of cropland extent develop by AFCD and LGRIP30 in Egypt and Nigeria, and background is the natural colour composite high-resolution imagery available in © Google Earth Engine.

## 5. Discussion

### 5.1 Cropland dynamics in Africa

Based on the produced AFCD map, we calculated country-level net changes in cropland area between 2000 and 2022, and we plotted annual cropland area dynamics at the continental scale. Overall, Africa's total cropland grew from 194.35 Mha in 2000 to 210.92 Mha in 2022 (Fig. 5c), a net increase of 16.57 Mha (8.53%). In terms of temporal changes, before 2005, the changes in arable land followed a wave-like pattern, with gradual fluctuations. However, after 2006, a sharp increase was observed (as shown in Fig. 9b). As seen in Fig. 9a, the overall arable land area in Africa shows a spatial increasing trend. Countries such as

the Democratic Republic of the Congo, Tanzania, and Mozambique have seen increases of over 1.5 million hectares (Mha), while most countries experienced an increase of less than 500,000 hectares (kha). For example, in the Democratic Republic of the Congo, 1,533.46 kha of land have been converted into arable land since the 21st century. In contrast, a few countries, including Zimbabwe (460.54 kha), South Sudan (88.70 kha), and Libya (0.25 kha), have seen a decline in arable land area.



This trend is linked to factors such as the growth of the mining industry in African nations, rural labor migration to mining

areas, or regional conflicts leading to the abandonment of arable land.

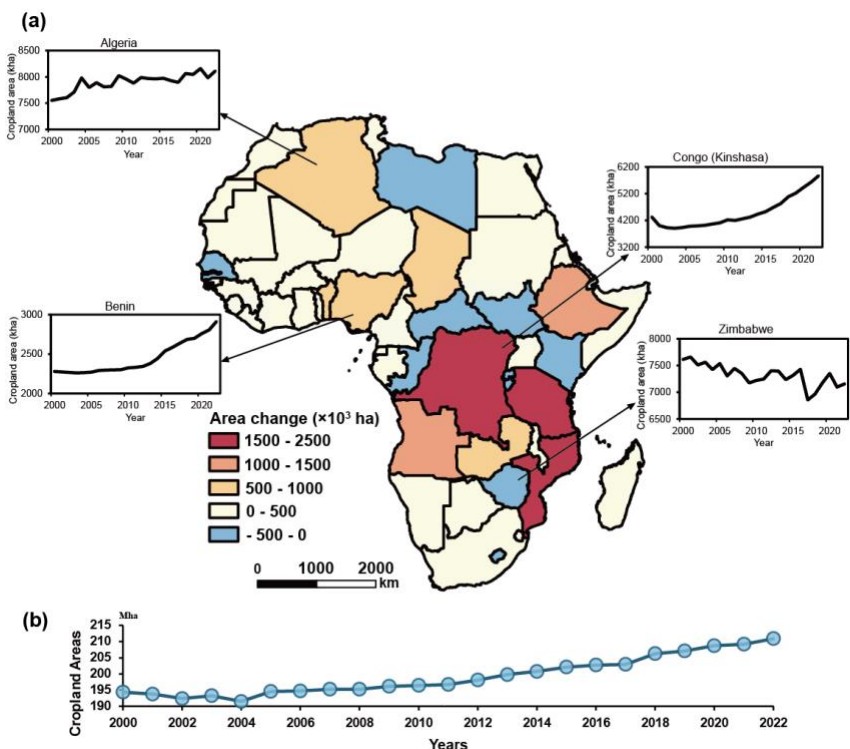

**Figure 9** Cropland area changes in Africa between 2000 and 2022. (a) Country-level cropland change patterns based on AFCD data; (b) Continent-wide annual cropland dynamics.

From 2000 to 2022, Africa's cropland area steadily increased; however, the area of abandoned cropland also rose. By 2018,

abandoned cropland accounted for 11.52% (24.70 Mha) of active cropland. In 2022, the increase in potentially abandoned cropland, projecting that abandoned cropland will constitute 13.31% of active cropland. The abandonment of cropland is driven by a combination of natural constraints, land degradation, demographic shifts, socio-economic factors, and institutional frameworks, which interact across different spatial and temporal scales (Zumkehr and Campbell, 2013). This study quantitatively evaluates the spatiotemporal trends and primary drivers of cropland abandonment in Africa. Countries situated

between 23°E and 39°E, including Egypt, Sudan, South Sudan, Uganda, Malawi, and Zimbabwe, account for 50.1% of the continent's abandoned cropland. Nationally, we found that most abandoned cropland remains unused for over five years. South Africa and Sudan exhibit the highest reclamation rates, with 10% of active cropland regenerated from previously abandoned areas, while most other nations have rates below 5%.

Cropland abandonment is influenced by a myriad of objective factors, encompassing agricultural conditions and the degree of

regional development. Furthermore, economic circumstances, demographic shifts, and urban expansion also significantly contribute to this phenomenon. A national-level analysis of cropland abandonment across Africa offers a spatial perspective

on the magnitude of abandonment, simultaneously reflecting the alterations in socioeconomic variables that underlie this trend. The heatmap reveals that countries such as Burundi, the Central African Republic, and Ghana exhibit both low areas and ratios of cropland abandonment (< 5%). Conversely, nations like Algeria, Burkina Faso, Mali, and Zambia consistently maintain higher levels of cropland abandonment annually, with abandoned areas exceeding 25% of active cropland. Furthermore, South Africa, Nigeria, Zimbabwe, and Sudan demonstrate elevated abandonment levels, notably with Nigeria experiencing significant abandonment around 2015.

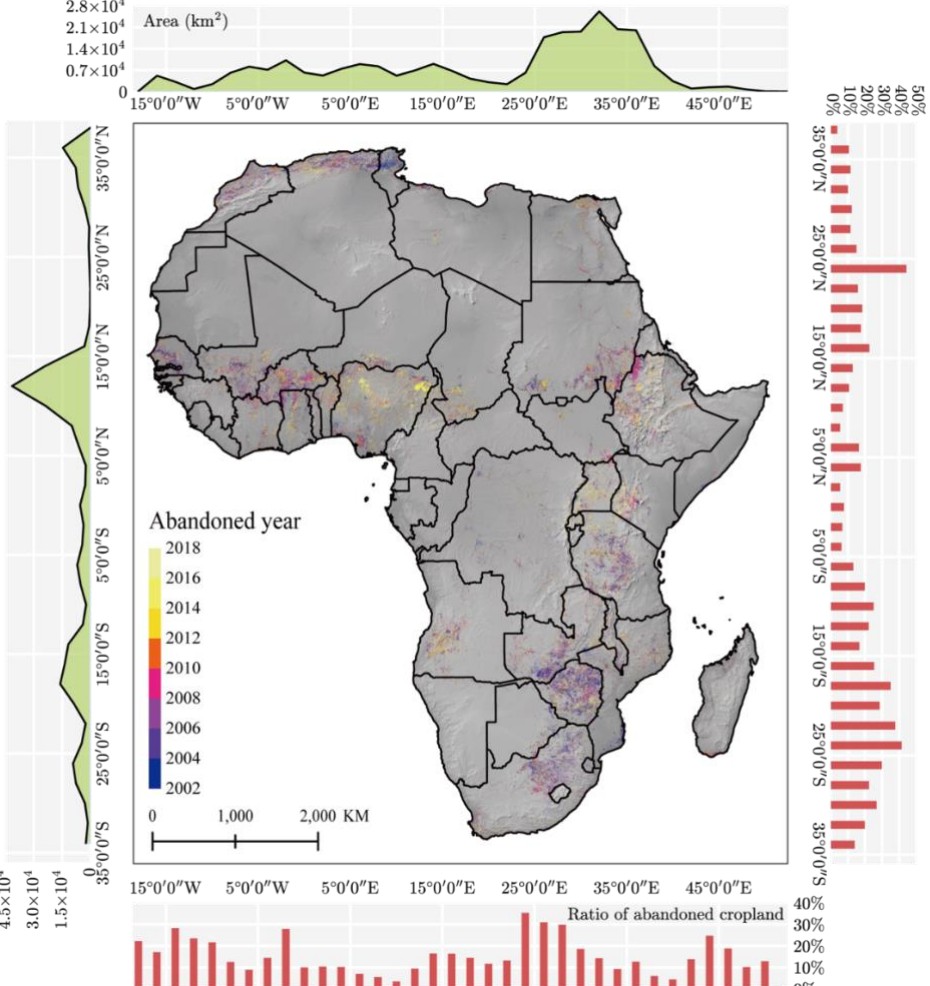

**Figure 10** Spatial distribution of cropland abandonment in Africa, indicating the year abandonment began. Includes a summary of abandoned cropland area and the ratio of abandoned to active cropland at 2° longitude and latitude intervals.

Cropland, a heavily modified landscape within the wildland-urban interface (WUI), is a critical source of carbon emissions due to agricultural biomass burning, which substantially influences climate change (Mallet et al., 2024). In Africa, rain-fed cropland predominates over irrigated farmland. As a result, farming activities are concentrated in the rainy season when water resources are abundant. During the dry season, arid conditions and natural vegetation growth create favorable conditions for



wildfires, driven by meteorological and fuel moisture dynamics. Studies indicate a global rise in wildfire risk, with Africa's WUI being the most affected (Chen et al., 2024). However, wildfire detection using MODIS sensors underestimates burned areas smaller than 100 ha by over 200% due to limited spatial resolution (Ramo et al., 2021). Agricultural fire management, commonly used by African farmers to clear crop residues and reduce wildfire risk during fallow periods (Laris et al., 2023; Ramo et al., 2021), is often absent on abandoned cropland, significantly heightening wildfire risks. This study's long-term

mapping of abandoned cropland in Africa offers a critical foundation for future research on fire management and monitoring wildfire risks in these areas.

Finally, cropland extraction in Africa is particularly challenging due to the complex crop planting structures in the region and the dynamic changes driven by climatic, environmental, and socio-economic factors. The long-term cropland data provided in this study is essential for understanding the interannual variation of cropland in Africa and its relationship to environmental,

climatic, land use, and policy influences.

## 5.2 The advantages of the AFCD map

This study presents an algorithm developed using Landsat time series data (2000-2022) implemented on GEE with Random Forest and CCDC, producing a 30-meter resolution cropland extent map for Africa (Fig. 5). By integrating existing LUCC products with the CCDC algorithm, we developed a consistent cropland map for Africa. This approach minimizes the impact

of seasonal false changes, ensuring stable and continuous identification of both active and abandoned croplands. Our method achieved an overall classification accuracy of 86% based on an independent validation dataset. Over the 22-year period, Africa experienced a net cropland expansion of 16.57 million hectares (Mha), representing nearly a 10% increase relative to the baseline levels at the start of the millennium. By comparing our product with other global and regional Land Use and Land Cover (LULC) or Cropland Extent (CE) products, we gain valuable insights into its strengths and limitations.

Existing LULC products, such as the ESRI 10m Annual Land Cover, GLAD's global cropland maps, and GlobeLand30, provide useful information; however, the AFCD product distinguishes itself through longer temporal coverage (2000–2022) and higher temporal frequency with Landsat-based annual observations at 30m resolution. This enables a more extended temporal observation of agricultural dynamics starting from the early 2000s. For example, compared to GLAD's quadrennial cropland extent product—aligned with FAO's five-year fallow land definition—the AFCD product offers enhanced temporal

resolution. This improvement significantly boosts detection capabilities for long-term land abandonment patterns. Additionally, the AFCD map reduces misclassifications of roads and buildings near farmland, a common issue in products like LGRIP.

When compared to other 30m-resolution products such as GLAD, GCEP, and LGRIP, the AFCD product shows good consistency in terms of total cropland area. However, it tends to slightly underestimate cropland areas when compared to higher-resolution products like the 10m-resolution DEA or the 300m-resolution ESACCI. As shown in Figure 8, while AFCD

preserves much of the object-level detail, the 30m resolution results in many mixed pixels, leading to small cropland areas being overlooked, especially when surrounded by vegetation.



A significant challenge in large-scale classification tasks is acquiring high-quality reference training and testing data. In this study, we generated a large number of sample data points and performed training and classification based on a grid approach. However, our training data relied on sample points derived from the limited overlap of different products. Cropland in sub-

Saharan Africa is often fragmented, forming mosaics with savannas, woodlands, and grasslands. As a result, various products may identify the same pixel containing cropland in different ways, leading to inconsistencies. Regions with low classification consistency across products are especially vulnerable to misclassification. Consequently, we included these challenging samples with limited recognition in our training dataset. While this introduces some uncertainty, it is a necessary trade-off to minimize the omission of cropland as much as possible.

The AFCD product facilitates: 1) continuous temporal tracking of non-linear cropland system evolution; 1) precise identification of abandonment onset years and associated trigger events; and 3) robust attribution analysis of climate and anthropogenic impacts on agricultural landscapes. Serving as both a historical record of 21st-century cropland dynamics in Africa and a foundational dataset for future abandonment studies, this product contributes significantly to continental food security initiatives, hunger mitigation strategies, and sustainable land management efforts.

Cropland mapping is a complex and dynamic process, marked by gradual or abrupt changes in land cover due to human activities. Previous studies have employed soft classification approaches alongside the LandTrendr algorithm to track cropland change trajectories (Dara et al., 2018; Xie et al., 2024). However, these methods rely heavily on the accuracy of cropland probability distribution maps, which can be affected by seasonal variations in image quality and difficulties in determining classification thresholds. Additionally, most research has focused on developed regions like Europe, China, and the United

States, leaving cropland change mapping in Africa, a region with diverse climates and complex land cover, a major challenge. The AFCD map provides more reliable information for cross-regional and cross-national comparisons and assessments. Such granular data is essential for understanding cropland transitions and guiding agricultural management practices in Africa.

## 5.3 The limitations and prospects of AFCD map

Our study acknowledges several limitations in terms of methodology and data. First, while we track cropland dynamics

annually, we do not account for intra-annual variations in crops, such as differences in crop calendars or planting intensities. As a result, our approach may not effectively capture perennial crops or multi-cropping systems. Second, we map the general extent of cropland without distinguishing between specific types, which is crucial when assessing crop yields or cropland's responses to climate and anthropogenic factors. Future research will likely benefit from incorporating advanced techniques for high-resolution crop type mapping. Third, while AFCD performs above the average accuracy level for similar products in

Africa, several factors contribute to its limitations. In terms of temporal accuracy, AFCD shows relatively lower precision in the early 21st century, mainly due to limited data availability for creating consensus label samples in the early 2000s, as well as uneven coverage of Landsat 5 data. Spatially, mapping accuracy varies by region, with relatively higher errors in countries near the Gulf of Guinea and in East Africa. This is due to local climate conditions and fragmented agricultural landscapes, which have long presented challenges for accurate cropland identification. Recent studies have highlighted the significant



discrepancies in cropland mapping across Africa (Kerner et al., 2024; Tubiello et al., 2023a). Additionally, the quality of Landsat data (e.g., unmasked cloud cover) and the sensitivity of the CCDC parameters have further impacted the mapping results. To improve cropland classification accuracy in these regions, future work should focus on comprehensive surveys and tailored strategies. Finally, it is important to note that the datasets used to generate consensus Crop/Non-crop samples define cropland differently, which may introduce noise and errors in our results. For instance, GLAD includes perennial herbaceous

plants, while GFSAD may encompass plantation lands such as those cultivated with fruits, coffee, and tea. By relying on consistency across products, our consensus labeling process partially excludes these areas. Lastly, although we assume that land cover types should not change between breakpoints identified by CCDC, significant breaks caused by human activities can alter the classification of cropland's duration of use. This can lead to overestimation or underestimation of cropland areas in certain regions in different years within the AFCD dataset.

## 6 Data availability

The developed AFCD map dataset can be freely accessed via https://doi.org/10.5281/zenodo.14920706 (Lou et al., 2025). To help users to navigate this dataset, it is saved as 23 independent files. Each file is named "AFCD_YYYY.tif", where YYYY is the year of cropland map. AFCD map contains 23 maps for time steps from 2000 to 2022, updated annually. In each TIFF file, a value of 1 indicates cropland area.

## 7 Conclusions

Cropland dynamics serve as a fundamental driver of human modification and adaptation to natural environments, while also constituting one of the ultimate objectives in Earth Observation research. Conducting large-scale and long-term cropland monitoring across Africa remains a significant challenge, particularly given the continent's complex agro-ecological conditions and persistent data limitations. This study generated an annual map of cropland extent at a 30-meter resolution for Africa

(namely AFCD), utilizing Landsat-5, Landsat-7, and Landsat-8 data spanning from 2002 to 2022. It highlighted the efficacy of mapping croplands over extensive areas by employing high-resolution satellite data processed through pixel-based RF algorithms and CCD algorithms on the GEE. The research introduced a novel methodology for sample generation, incorporating crop and non-crop samples from previous LUCC and cropland extent products, alongside RF and CCD algorithms, to produce annual maps of cropland extent in Africa. An independent accuracy assessment revealed an overall map

accuracy of 0.86. By 2022, Africa's cropland expanded from with a net increase of 8.53%. The high spatial and temporal resolution of the maps enabled detailed capture of cropland change across Africa, helping to uncover the impact of cropland on the climate, improve food security, and develop sustainable land management practices.



**Author contributions.** DP and ZL conceived the research idea. ZL designed the study, created the dataset, carried out the
analysis, and wrote the first draft of the manuscript, under the supervision of other authors. All authors participated in the
review and editing of the manuscript.

**Competing interests.** The contact author has declared that none of the authors has any competing interests.

**Acknowledgements.** The authors express their great gratitude to the following organizations for their contributions to this
research: USGS for providing free access to Landsat imagery and GFSAD data, the University of Maryland for providing
GLAD data, the Aerospace information research institute for providing GLC_FCS30D, the ESA for providing WorldCover,
Dynamic World and CCI Land cover data, the Esri for providing 10m annual land cover data (2017 - 2023), the Boston
University for providing GLanCE samples and Geo-Wiki for sharing the third-party validation samples

**Financial support.** This study was supported by the National Key R&D Program of China (grant no. 2023YFE0207900).

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
