# Peer review of "An Annual Cropland Extent Dataset for Africa at 30m Spatial Resolution from 2000 to 2022"

_Earth System Science Data, 2025_

## Author Comment (AC1)

**Response to Referee #1**

We appreciate you very much for your comments concerning our manuscript entitled "A 30m resolution annual cropland extent dataset of Africa in recent decades of the 21st century" (MS No.: ESSD-2025-133). Those comments are valuable and helpful for improving our manuscript. We followed all comments and made revision and responses carefully. Revised portions are marked in *Orange* in the revised manuscript. The line, and figure numbers refer to our revised manuscript. And, a point-by-point reply to the comments are listed below.

Q1: Title: Consider this title instead – An Annual Cropland Extent Dataset for Africa at 30m Spatial Resolution from 2000 to 2022

A1: Thank you very much for your valuable suggestion regarding the title. We fully agree that the proposed title more clearly conveys the scope and content of the study. Accordingly, we have revised the title to "*An Annual Cropland Extent Dataset for Africa at 30m Spatial Resolution from 2000 to 2022*", as reflected on Page 1 of the revised manuscript.

Q2: Abstract, line 25: "The study developed a 30-meter resolution African annual cropland distribution (namely AFCD) dataset for Africa spanning the years 2000 to 2022."— delete "for Africa".

A2: Thank you for pointing this out. We agree that the phrase "for Africa" is redundant in this context, as the geographical scope is already clear from the rest of the sentence. We have removed "for Africa" from the abstract accordingly (Line 25 of the revised manuscript).

Q3: Introduction, Line 40: change "croplands play is of critical importance" to "croplands are of critical importance"

A3: Thank you for your careful reading and suggestion. We have corrected the sentence in Line 41 of the Introduction to "croplands are of critical importance" to improve grammatical accuracy and clarity.

Q5: Line 30-31 mentioned results of R-square to compare with other product, if it is a part of evaluation accuracy assessment maybe worth to include it in 3.4 Accuracy assessment as well

A5: Thank you for your insightful suggestion. We agree that including the R-square results as part of the accuracy assessment would enhance the clarity and completeness of the evaluation. In response, we have added the equation for R-square in Section 3.4 (Lines 240–244) to enhance transparency and reproducibility.

"... complementary approaches. First, we evaluated the accuracy of AFCD by comparing its national-scale area statistics with official FAO reports and other existing cropland products, using R-square ( $R^2$ , shown as Eq. 2) as the evaluation metric. Second, we performed sample-based ...

$$R^{2} = 1 - \frac{\sum (y_{i} - \hat{y}_{i})^{2}}{\sum (y_{i} - \bar{y}_{i})^{2}}$$
(2)"

**Q6: Line 36: what does LGRIP stand for?**

A6: Thank you for your comment. After consideration, we decided to remove the mention of LGRIP in the abstract. We revised the sentence in Line 34 to:

"*AFCD also avoided the misclassification of buildings, roads, and trees surrounding cropland, common in existing products.*"

Q7: Line 76: what does GCEP stand for? Also is it a mistake in the citation? the cite 'Xiong et al., 2017b' refers to dataset Global Food Security-support Analysis Data (GFSAD) not GCEP

A7: Thank you very much for pointing out this issue. To avoid confusion, we have now provided the full names for both GCEP and LGRIP at their first mentions. In addition, regarding the citation, the reference "Xiong et al., 2017b" was mistakenly cited as the source for GCEP. After careful verification, we have corrected this to the appropriate reference:

Thenkabail, P. S., Teluguntla, P. G., Xiong, J., Oliphant, A., Congalton, R. G., Ozdogan, M., Gumma, M. K., Tilton, J. C., Giri, C., Milesi, C., Phalke, A., Massey, R., Yadav, K., Sankey, T., Zhong, Y., Aneece, I., and Foley, D.: Global cropland-extent product at 30-m resolution (GCEP30) derived from landsat satellite time-series data for the year 2015 using multiple machine-learning

algorithms on Google earth engine cloud, Professional Paper, U.S. Geological Survey, https://doi.org/10.3133/pp1868, 2021.

Accordingly, the sentence has been revised as follows:

"While specialized cropland products, such as the Landsat Global Cropland Extent (Potapov et al., 2022), GFSAD Landsat-Derived Global Rainfed and Irrigated-Cropland Product (Teluguntla et al., 2023), GFSAD Global Cropland Extent Product (Thenkabail et al., 2021), and Digital Earth Africa..."

Q8: Mention table 1 somewhere in L101-103 could make the context more clear and help to understand all the dataset names in the following paragraph

A8: Thank you for the constructive suggestion. To improve clarity and help readers better understand the dataset names referenced in the following paragraph, we have added a reference to Table 1 in Lines 105. The revised sentence now reads:

"To investigate the spatial distribution of Africa within the study's temporal scope, we amassed a comprehensive dataset from Google Earth Engine (GEE) and its community platform encompassing global or Africa-specific land use/cover change (LUCC) datasets as well as cropland distribution, with a spatial resolution finer than 500 meters, spanning the period from 2000 to 2022 (as summarized in Table 1)."

Q9: Should the Fig 1 (a) -(e) be mentioned somewhere? If the plan is to only mention (f) in L197 maybe just make it an individual figure. Also it's quite confusing what is the relationship between (a) - (e) and (f), there are two similar color bar for different meaning

A9: Thank you for your insightful comment. Figure 1 was designed to provide essential background context for understanding cropland mapping in Africa. Subplots (a)–(e) display the spatial distribution of valid Landsat observations over five successive periods (2000–2002, 2003–2007, 2008–2012, 2013–2017, and 2018–2022), which helps assess the temporal density and data availability across regions. This is particularly important because the quality and density of Landsat observations have a significant influence on the accuracy of cropland mapping.

Subplot (f), on the other hand, presents a cropland consensus map derived from six different LULC/cropland products, with values from 1 to 6 indicating the number of products that agree on a pixel being classified as cropland. This layer was used to guide the construction of  $2^{\circ} \times 2^{\circ}$  grid zones for sampling and analysis across Africa.

We have updated the text in Line 133-139 to clarify this comment:

"As shown in Figure 1a–e, the spatial distribution of valid Landsat observations over five time periods exhibits significant spatiotemporal variability, which can directly affect the accuracy of cropland mapping. To mitigate this influence and enhance spatial consistency, we incorporated a cropland consensus layer derived from six global LULC/cropland products, where cropland agreement levels are categorized from SA1 to SA6 (Figure 1f) (Tubiello et al., 2023a, 2023b). This consensus informed the construction of a  $2^{\circ} \times 2^{\circ}$  grid framework, delineated by red lines in Figure 1f, with each grid cell served as the minimum classification unit for the cropland mapping process."

Then we have revised the caption of Figure 1 to further clarify the color bar of subplot (f):

"Figure 1 Temporal and Spatial Distribution of Available Landsat Imagery Data Over Africa and (a-e) and Spatial Distribution of Cropland derived from six global LULC/cropland products (f). In (f), SA1–SA6 represent increasing cropland agreement levels (Tubiello et al., 2023a, 2023b)."

Q10: For Fig 2 (b), a subplot showing location (which part of Africa) and size of the area could be helpful for the reader to Also if the plan is to mention (b) before (a), the sequence can be switched.

A10: Thank you for your thoughtful suggestion. As advised, we have switched the order of the original subplots in Figure 2 (Figure R1) to ensure consistency with the sequence in which they are referenced in the text. In addition, to help readers better understand the spatial context of the selected area, we have added a subplot to the updated Figure 2(a) showing the location and spatial extent of the example area within Africa.

We have modified the sentences in Line 133 and Line 135, and the Figure 2 has been updated:

"... surveys and visual interpretation across eight countries in sub-Saharan Africa (shown as Fig. 2 (a)). Additionally, we utilized crowdsourced data provided by (Laso Bayas et al., 2017), collected via the Geo-Wiki platform, for validation (Fig. 2 (b)). The dataset selected ..."

Figure R2 Spatial distribution of validation points for regional and time-series.

Q11: Line 136 mentioned 'samples were randomly selected for further validation by students and experts': quite confused about what kind of validation has been done here, brief introduction about the validation method and validation result could be better

A11: Thank you for your valuable comment. We agree that the original description was unclear, and the cropland sample data used in this study were derived from a crowdsourcing campaign conducted on the Geo-Wiki platform. Participants visually interpreted whether cropland was present at selected locations and estimated its proportional area within a 300 m  $\times$  300 m grid. To ensure data quality, a subset of samples was randomly selected for secondary validation by three trained students. Samples with inconsistent results across the three students were excluded. In addition, experts further reviewed and validated these control samples to improve the overall reliability of the dataset.

We have updated the text in Line 135-140 to clarify this comment:

"The dataset includes 35,866 cropland samples collected globally through a crowdsourcing campaign on the Geo-Wiki platform, with 7,313 samples located in Africa. Each sample corresponds to a 300 m  $\times$  300 m PROBA-V grid cell and records the proportion of cropland within the frame, based on visual interpretation by participants. To ensure data quality, a subset of samples was randomly selected for secondary validation by three trained students (provide as control samples). Samples with inconsistent results among the student interpretations were

excluded. Additionally, experts further reviewed and validated the control samples to enhance the overall reliability of the dataset (provide as expert samples) (Laso Bayas et al., 2017)."

**Q12: Fig 2 in L142 is a typo?**

A12: Thank you for pointing this out. We confirm that the reference to "Fig 2" in Line 142 was a typographical error. It should have referred to "Figure 3," and we have corrected this in the revised manuscript accordingly.

Q13: In section 3.1, the step of reclassification existing LULC into the cropland/ non-cropland under your definition (the 'remapping' in L160) should be mentioned; it can better explain how you make the different LULC definition consistent for your use.

A13: Thank you for your suggestion. We have updated Table R1 to include the cropland definitions and class label numbers used in each LULC/cropland product. Based on these definitions, we reclassified the original classes into cropland and non-cropland.

| Dataset                       | Year(s)                            | Res.
(m/px) | Coverage | Definition of cropland                                                                                                                                                                                                                                         | Cropland
class no. |
|-------------------------------|------------------------------------|----------------|----------|----------------------------------------------------------------------------------------------------------------------------------------------------------------------------------------------------------------------------------------------------------------|-----------------------|
| GLC_FCS30D                    | 1985; 1990;
1995; 2000-
2022 | 30m            | Global   | Irrigated cropland, Rainfed cropland,
Herbaceous cover cropland, Tree or
shrub cover cropland (Zhang et al.,
2024).                                                                                                                                   | 10, 11, 12, 20        |
| ESRI 10m Annual
Land Cover | 2017-2023                          | 10m            | Global   | Crops Human planted/plotted
cereals, grasses, and crops not at tree
height; examples: corn, wheat, soy,
fallow plots of structured land (Karra
et al., 2021).                                                                                      | 5                     |
| ESA WorldCover                | 2020; 2021                         | 10m            | Global   | Land covered with annual cropland
that is sowed/planted and harvestable
at least once within the 12 months
after the sowing/planting date. The
annual cropland produces a
herbaceous cover and is sometimes
combined with some tree or woody | 40                    |

Table R1 Description of map data products utilized in this study.

|                                                                                                                        |                                                                                       |                           |                                                   | vegetation. Note that perennial                                                                                                                                                                                                                                                                                                                                                                                                                                                                                                                                                                                                                                                                                                                                                                                                                                                                                                                                                                    |                                                                           |
|------------------------------------------------------------------------------------------------------------------------|---------------------------------------------------------------------------------------|---------------------------|---------------------------------------------------|----------------------------------------------------------------------------------------------------------------------------------------------------------------------------------------------------------------------------------------------------------------------------------------------------------------------------------------------------------------------------------------------------------------------------------------------------------------------------------------------------------------------------------------------------------------------------------------------------------------------------------------------------------------------------------------------------------------------------------------------------------------------------------------------------------------------------------------------------------------------------------------------------------------------------------------------------------------------------------------------------|---------------------------------------------------------------------------|
|                                                                                                                        |                                                                                       |                           |                                                   | woody crops will be classified as the                                                                                                                                                                                                                                                                                                                                                                                                                                                                                                                                                                                                                                                                                                                                                                                                                                                                                                                                                              |                                                                           |
|                                                                                                                        |                                                                                       |                           |                                                   | appropriate tree cover or shrub land                                                                                                                                                                                                                                                                                                                                                                                                                                                                                                                                                                                                                                                                                                                                                                                                                                                                                                                                                               |                                                                           |
|                                                                                                                        |                                                                                       |                           |                                                   | cover type. Greenhouses are                                                                                                                                                                                                                                                                                                                                                                                                                                                                                                                                                                                                                                                                                                                                                                                                                                                                                                                                                                        |                                                                           |
|                                                                                                                        |                                                                                       |                           |                                                   | considered as built-up (Zanaga et al.,                                                                                                                                                                                                                                                                                                                                                                                                                                                                                                                                                                                                                                                                                                                                                                                                                                                                                                                                                             |                                                                           |
|                                                                                                                        |                                                                                       |                           |                                                   | 2022, 2021).                                                                                                                                                                                                                                                                                                                                                                                                                                                                                                                                                                                                                                                                                                                                                                                                                                                                                                                                                                                       |                                                                           |
|                                                                                                                        |                                                                                       |                           |                                                   | Estimated probability of complete                                                                                                                                                                                                                                                                                                                                                                                                                                                                                                                                                                                                                                                                                                                                                                                                                                                                                                                                                                  |                                                                           |
| Dynamic World                                                                                                          | 2015-now                                                                              | 10m                       | Global                                            | coverage by crops(Brown et al.,                                                                                                                                                                                                                                                                                                                                                                                                                                                                                                                                                                                                                                                                                                                                                                                                                                                                                                                                                                    | 4                                                                         |
|                                                                                                                        |                                                                                       |                           |                                                   | 2022).                                                                                                                                                                                                                                                                                                                                                                                                                                                                                                                                                                                                                                                                                                                                                                                                                                                                                                                                                                                             |                                                                           |
|                                                                                                                        |                                                                                       |                           |                                                   | Rainfed cropland, irrigated or post-                                                                                                                                                                                                                                                                                                                                                                                                                                                                                                                                                                                                                                                                                                                                                                                                                                                                                                                                                               |                                                                           |
|                                                                                                                        |                                                                                       |                           |                                                   | flooding cropland, Mosaic cropland                                                                                                                                                                                                                                                                                                                                                                                                                                                                                                                                                                                                                                                                                                                                                                                                                                                                                                                                                                 |                                                                           |
| ESA CCI                                                                                                                | 1992-2020                                                                             | 300m                      | Global                                            | (>50%) / natural vegetation (tree,                                                                                                                                                                                                                                                                                                                                                                                                                                                                                                                                                                                                                                                                                                                                                                                                                                                                                                                                                                 | 10, 20, 30                                                                |
|                                                                                                                        |                                                                                       |                           |                                                   | shrub, herbaceous cover) (<50%)                                                                                                                                                                                                                                                                                                                                                                                                                                                                                                                                                                                                                                                                                                                                                                                                                                                                                                                                                                    |                                                                           |
|                                                                                                                        |                                                                                       |                           |                                                   | (ESA, 2017).                                                                                                                                                                                                                                                                                                                                                                                                                                                                                                                                                                                                                                                                                                                                                                                                                                                                                                                                                                                       |                                                                           |
|                                                                                                                        |                                                                                       |                           |                                                   | Sowed/planted and harvestable at                                                                                                                                                                                                                                                                                                                                                                                                                                                                                                                                                                                                                                                                                                                                                                                                                                                                                                                                                                   |                                                                           |
| Digital Earth Africa                                                                                                   |                                                                                       | 10m                       | Continent                                         | least once within the 12 months after                                                                                                                                                                                                                                                                                                                                                                                                                                                                                                                                                                                                                                                                                                                                                                                                                                                                                                                                                              | 1                                                                         |
| Cropland Extent                                                                                                        | 2019                                                                                  |                           |                                                   | the sowing/planting date (Burton et                                                                                                                                                                                                                                                                                                                                                                                                                                                                                                                                                                                                                                                                                                                                                                                                                                                                                                                                                                |                                                                           |
|                                                                                                                        |                                                                                       |                           |                                                   | al., 2022).                                                                                                                                                                                                                                                                                                                                                                                                                                                                                                                                                                                                                                                                                                                                                                                                                                                                                                                                                                                        |                                                                           |
| GFSAD global
cropland maps                                                                                          | 2015                                                                                  | 30m                       | Global                                            | Rainfed cropland (cropland areas                                                                                                                                                                                                                                                                                                                                                                                                                                                                                                                                                                                                                                                                                                                                                                                                                                                                                                                                                                   | 1, 2                                                                      |
|                                                                                                                        |                                                                                       |                           |                                                   | that are purely dependent on direct                                                                                                                                                                                                                                                                                                                                                                                                                                                                                                                                                                                                                                                                                                                                                                                                                                                                                                                                                                |                                                                           |
|                                                                                                                        |                                                                                       |                           |                                                   | precipitation), irrigated cropland                                                                                                                                                                                                                                                                                                                                                                                                                                                                                                                                                                                                                                                                                                                                                                                                                                                                                                                                                                 |                                                                           |
|                                                                                                                        |                                                                                       |                           |                                                   | (cropland that had at least one                                                                                                                                                                                                                                                                                                                                                                                                                                                                                                                                                                                                                                                                                                                                                                                                                                                                                                                                                                    |                                                                           |
|                                                                                                                        |                                                                                       |                           |                                                   | irrigation during the crop growing                                                                                                                                                                                                                                                                                                                                                                                                                                                                                                                                                                                                                                                                                                                                                                                                                                                                                                                                                                 |                                                                           |
|                                                                                                                        |                                                                                       |                           |                                                   | period) (Thenkabail et al., 2021).                                                                                                                                                                                                                                                                                                                                                                                                                                                                                                                                                                                                                                                                                                                                                                                                                                                                                                                                                                 |                                                                           |
|                                                                                                                        |                                                                                       |                           |                                                   | Land used for annual and perennial                                                                                                                                                                                                                                                                                                                                                                                                                                                                                                                                                                                                                                                                                                                                                                                                                                                                                                                                                                 |                                                                           |
| GLAD Global
Cropland Maps                                                                                           | 2003, 2007,
2011, 2015,
2019                                                    | 30m                       | Global                                            | herbaceous crops for human                                                                                                                                                                                                                                                                                                                                                                                                                                                                                                                                                                                                                                                                                                                                                                                                                                                                                                                                                                         | 1                                                                         |
|                                                                                                                        |                                                                                       |                           |                                                   | consumption, forage (including hay)                                                                                                                                                                                                                                                                                                                                                                                                                                                                                                                                                                                                                                                                                                                                                                                                                                                                                                                                                                |                                                                           |
|                                                                                                                        |                                                                                       |                           |                                                   | and biofuel. Perennial woody crops,                                                                                                                                                                                                                                                                                                                                                                                                                                                                                                                                                                                                                                                                                                                                                                                                                                                                                                                                                                |                                                                           |
|                                                                                                                        |                                                                                       |                           |                                                   | permanent pastures and shifting                                                                                                                                                                                                                                                                                                                                                                                                                                                                                                                                                                                                                                                                                                                                                                                                                                                                                                                                                                    |                                                                           |
|                                                                                                                        |                                                                                       |                           |                                                   | cultivation are excluded from the                                                                                                                                                                                                                                                                                                                                                                                                                                                                                                                                                                                                                                                                                                                                                                                                                                                                                                                                                                  |                                                                           |
|                                                                                                                        |                                                                                       |                           |                                                   | definition. The fallow length is                                                                                                                                                                                                                                                                                                                                                                                                                                                                                                                                                                                                                                                                                                                                                                                                                                                                                                                                                                   |                                                                           |
|                                                                                                                        |                                                                                       |                           |                                                   | limited to 4 years for the cropland                                                                                                                                                                                                                                                                                                                                                                                                                                                                                                                                                                                                                                                                                                                                                                                                                                                                                                                                                                |                                                                           |
|                                                                                                                        |                                                                                       |                           |                                                   | class (Potapov et al., 2022).                                                                                                                                                                                                                                                                                                                                                                                                                                                                                                                                                                                                                                                                                                                                                                                                                                                                                                                                                                      |                                                                           |
| Africa Cropland
Mask                                                                                                | 2016                                                                                  | 30m                       | Continent                                         | Agricultural annual standing                                                                                                                                                                                                                                                                                                                                                                                                                                                                                                                                                                                                                                                                                                                                                                                                                                                                                                                                                                       | 1                                                                         |
|                                                                                                                        |                                                                                       |                           |                                                   | croplands, cropland fallows, and                                                                                                                                                                                                                                                                                                                                                                                                                                                                                                                                                                                                                                                                                                                                                                                                                                                                                                                                                                   |                                                                           |
|                                                                                                                        |                                                                                       |                           |                                                   | permanent plantation crops (Nabil et                                                                                                                                                                                                                                                                                                                                                                                                                                                                                                                                                                                                                                                                                                                                                                                                                                                                                                                                                               |                                                                           |
|                                                                                                                        |                                                                                       |                           |                                                   | al., 2022).                                                                                                                                                                                                                                                                                                                                                                                                                                                                                                                                                                                                                                                                                                                                                                                                                                                                                                                                                                                        |                                                                           |
| ESA CCI Digital Earth Africa Cropland Extent GFSAD global cropland maps GLAD Global Cropland Maps Africa Cropland Mask | 1992-2020         2019         2015         2003, 2007, 2011, 2015, 2019         2016 | 300m
10m
30m
30m | Global
Global
Global
Global
Continent | 2022).
Rainfed cropland, irrigated or post-
flooding cropland, Mosaic cropland
(>50%) / natural vegetation (tree,
shrub, herbaceous cover) (<50%)
(ESA, 2017).
Sowed/planted and harvestable at
least once within the 12 months after
the sowing/planting date (Burton et
al., 2022).
Rainfed cropland (cropland areas
that are purely dependent on direct
precipitation), irrigated cropland
(cropland that had at least one
irrigation during the crop growing
period) (Thenkabail et al., 2021).
Land used for annual and perennial
herbaceous crops for human
consumption, forage (including hay)
and biofuel. Perennial woody crops,
permanent pastures and shifting
cultivation are excluded from the
definition. The fallow length is
limited to 4 years for the cropland
class (Potapov et al., 2022).
Agricultural annual standing
croplands, cropland fallows, and
permanent plantation crops (Nabil et | 10, 20, 30         1         1         1, 2         1         1         1 |

And we updated the text in second step of reference samples generation:

"Based on the cropland definitions and class label numbers in Table 1, each LULC product was reclassified into cropland and non-cropland (excluding categories with significant ..."

Q14: In Line 195, 'Previous studies have shown that the classification accuracy of this algorithm is not significantly affected by the specific parameter values.' maybe worth citing which studies

A14: Thank you for your helpful comment. We agree that providing a citation strengthens the credibility of this statement. Accordingly, we have added the references to the relevant study in Line 199 of the revised manuscript to support the claim regarding the algorithm's robustness to parameter variations.

We have updated the text in Line 199 to clarify this comment:

"Previous studies have shown that the classification accuracy of this algorithm is not significantly affected by the specific parameter values (Belgiu, 2016; Zhang et al., 2019)."

**Reference:**

Belgiu, M.: Random forest in remote sensing: a review of applications and future directions, ISPRS J. Photogramm. Remote Sens., 114, 24–31, https://doi.org/10.1016/j.isprsjprs.2016.01.011, 2016. Zhang, X., Liu, L., Chen, X., Xie, S., and Gao, Y.: Fine land-cover mapping in China using landsat datacube and an operational SPECLib-based approach, Remote Sens., 11, 1056, https://doi.org/10.3390/rs11091056, 2019.

Q15: In Line 208, 'the algorithm first decomposes the time-series data into trend and periodic components using the Fourier transform' and then? feel some paragraph is missing here for how you capture changes

A15: Thank you for pointing this out. We agree that the original explanation was incomplete. To clarify how changes are detected by the algorithm, we have revised the description in Line 208:

"This algorithm employs Fourier transform techniques to model time-series observations using trend terms (to estimate trend changes) and harmonic terms (to describe periodic changes), then identifies potential land use disruptions when the time-series reflectance data exhibits six consecutive outliers that significantly deviate from the model-fitted curve (Zhu and Woodcock, 2014). This detection criterion typically corresponds to substantial surface modifications,

including but not limited to afforestation initiatives (Decuyper et al., 2022), agricultural reclamation projects (Chen et al., 2023), or spontaneous cropland abandonment, thereby facilitating precise detection of changes in cropland regions."

This addition helps to better illustrate the underlying change detection mechanism.

Q16: What is m and what is N in Equation (2)? - I assume m is category number and N is sample number but you need to specify here; eg. if your binary category is 0 and 1, the m should be [0,1] but in that case the k should start 0

A16: Thank you for your careful observation. We agree that the definitions of m and N were unclear and could cause confusion. In the revised manuscript, we have clarified that m represents the number of classes (which is 2 in our binary classification: cropland vs. non-cropland), and N denotes the total number of validation samples. Additionally, we have corrected the summation in Equation (3) so that k starts from 0 and iterates up to m - 1, consistent with the class labels.

We have modified sentences in Line 244:

"...matrix analysis, as shown in Eq. (3):

$$PA_{k} = \frac{p_{kk}}{p_{k}}$$

$$UA_{k} = \frac{p_{kk}}{p_{\cdot k}}$$

$$OA = \frac{\sum_{k=0}^{m-1} p_{kk}}{N}$$

$$F1_{k} = \frac{2 \cdot PA_{k} \cdot UA_{k}}{PA_{k} + UA_{k}}$$

$$Kappa = \frac{N \sum_{k=0}^{m-1} p_{kk} - \sum_{k=0}^{m-1} p_{k} \cdot p_{\cdot k}}{N^{2} - \sum_{k=0}^{m-1} p_{k} \cdot p_{\cdot k}}$$
(3)

For binary classification (cropland vs. non-cropland), we define the number of classes as m = 2, corresponding to class labels 0 and 1. Let N be the total number of validation samples and  $p_{ij}$  denote the ..."

Q17: You mentioned 'similarity matrices' in L245 but it is hard to understand with simply a cite. Similarity matrix is the confusion matrix in the paper you cited, which is the same thing you used for OA F1 calculation, not something that paper created. If you intended to express you used it like Phalke et al., 2020 did, briefly mention how it can be better with just a cite. A17: Thank you for the helpful comment. We acknowledge that the term "similarity matrices" may be unclear without context. In our study, we constructed confusion matrices to compare our results with other land cover products, named as similarity matrices like Phalke et al. (2020).

We have updated the manuscript accordingly to improve clarity (Line 277).

"... with other remote sensing-based land cover products. From these matrices, we computed standard accuracy metrics including PA, UA, F1-score, and overall similarity (equivalent in calculation to OA). It should be specifically noted... "

Q18: In Table 2 (L280), why not all the highest values are highlighted? Eg. Accuracy and Precision for Mali

A18: Thank you for your observation. In Table 2, we followed a consistent highlighting scheme: the highest value in each row is highlighted in bold blue, and the second highest in bold black. We have double-checked the table to ensure that this rule has been strictly applied across all metrics, including the values for Mali. We appreciate your attention to detail, which helped us further verify the consistency of our formatting.

Q19: What do 'control samples' and 'expert samples' refers to in L285 and L286

A19: Thank you for your question. The terms "control samples" and "expert samples" refer to the subset of cropland validation samples used to assess and ensure data quality within the Geo-Wiki crowdsourcing campaign. As described in the revised Line 138 and in our response to Q15, control samples were selected from the full dataset and further validated by three trained students to ensure internal consistency. "Expert samples" refer to those same control samples that were additionally reviewed and confirmed by experts to provide a high-confidence reference set.

Q20: Some details for plot explanation could be added in caption of Figure 6, eg. what is the red/blue point, what is the purple line etc

A20: Thank you for your helpful suggestion. We have revised the caption of Figure 6 to include more detailed explanations of the plot elements. Specifically, in subplots (b–f), the **blue** and **red points** represent the differences between AFCD and other LULC/Cropland products with respect to FAO data, respectively. The **purple line** indicates the linear fit between AFCD and the other products. These additions aim to help readers better interpret the figure.

We have updated the caption of Figure 6 (Line 317) to enhance the clarity and readability of the figure:

"Figure 3 Comparison of AFCD with FAO Statistical Cropland Area (a) and Other LULC/Cropland Products (b-f) (2000–2022). In (b–f), blue and red dots show the differences from FAO estimates for AFCD and other products, respectively. The purple line indicates the linear fit between AFCD and each product."

Q21: The discussion of advantages and limitations is objective and fair.

A21: Thank you for your positive comment.

---

## Author Comment (AC2)

**Response to Referee #2**

We appreciate you very much for your comments concerning our manuscript entitled "A 30m resolution annual cropland extent dataset of Africa in recent decades of the 21st century" (MS No.: ESSD-2025-133). Those comments are valuable and helpful for improving our manuscript. We followed all comments and made revision and responses carefully. Revised portions are marked in *Orange* in the revised manuscript. The line, and figure numbers refer to our revised manuscript. And, a point-by-point reply to the comments are listed below.

Major concerns:

Q1. The training data incorporates multiple existing products, yet inconsistencies exist in their cropland definitions. How did the authors address the noise introduced by such definitional discrepancies?

A1: Thank you for this insightful question. To address the inconsistencies in cropland definitions across different products used for training, we first summarized the cropland definitions and corresponding class label numbers for each product in the revised Table 1. In our training data generation, the operational definition of cropland was closely aligned with the source products. Specifically, our cropland definition encompasses both rainfed and irrigated systems and includes cropland-dominated agroforestry systems with mixed vegetation, while excluding perennial woody plantations and greenhouse-covered lands. To further reduce noise and ensure temporal consistency, we incorporated the Continuous Change Detection (CCD) algorithm to help distinguish cropland from natural vegetation or abandoned land. Overall, our cropland product aligns with the FAO's general definition of arable land and temporary crops, while also reflecting the smallholder-dominated, mixed-use farming landscapes typical of African agricultural systems.

Q2. The Discussion section mentions the utilization of AFCD data for spatial mapping of abandoned cropland in Africa, which represents a highly meaningful endeavour. However, it should be noted that the authors did not specify how abandoned land was defined in this study. We recommend that the authors incorporate relevant descriptions regarding the operational definition of abandoned cropland.

A1: Thank you for your constructive comment. We agree that the definition of abandoned cropland should be clearly stated. In response, we have added a description to the manuscript clarifying that, following the FAO definition, land is considered abandoned when previously cultivated cropland remains idle for more than five consecutive years.

We have updated the text in Line 391 to clarify this definition:

"*… the area of abandoned cropland also rose. According to the FAO, cropland abandonment refers to formerly cultivated land that has not been used for agricultural production for a period exceeding five consecutive years. By 2018, abandoned cropland …*"

Q3. The Geo-Wiki sample (Laso Bayas et al., 2017) is based on 300m PROBA-V imagery, but AFCD is a 30m product. Does this scale difference lead to validation bias?

A3: Thank you for your insightful comment. Although our AFCD product has a spatial resolution of 30 m, the Geo-Wiki cropland dataset provided by Laso Bayas et al. (2017) accounts for scale compatibility by subdividing each 300 m × 300 m sample into a 10 × 10 grid of 30 m cells. These cells, matching the resolution of our product, were visually interpreted by multiple participants. The final cropland proportion per 300 m grid was aggregated from these 30 m interpretations, with the median value across at least three independent annotators used to reduce individual subjectivity. During validation, we aggregated our product to the 300 m scale to align with the reference data.

This approach is consistent with established validation practices. Notably, Waldner et al. (2019) demonstrated the applicability and reliability of the Geo-Wiki dataset for validating 30 m binary cropland maps such as GFSAD and GlobeLand30. Similarly, Nabil et al. (2020) employed a comparable method by estimating cropland percentages within 300 m × 300 m areas based on counts of high-resolution cropland pixels. Given that the Geo-Wiki reference data originates from 30 m-level interpretation and has been validated in previous studies, and considering our use of resolution-aligned aggregation, we believe that the scale difference does not introduce significant bias in our validation.

Reference:
Laso Bayas, J. C., Lesiv, M., Waldner, F., Schucknecht, A., Duerauer, M., See, L., Fritz, S., Fraisl, D., Moorthy, I., McCallum, I., Perger, C., Danylo, O., Defourny, P., Gallego, J., Gilliams, S., Akhtar, I. ul H., Baishya, S. J.,

Baruah, M., Bungnamei, K., Campos, A., Changkakati, T., Cipriani, A., Das, K., Das, K., Das, I., Davis, K. F., Hazarika, P., Johnson, B. A., Malek, Z., Molinari, M. E., Panging, K., Pawe, C. K., Pérez-Hoyos, A., Sahariah, P. K., Sahariah, D., Saikia, A., Saikia, M., Schlesinger, P., Seidacaru, E., Singha, K., and Wilson, J. W.: A global reference database of crowdsourced cropland data collected using the geo-wiki platform, Sci. Data, 4, 170136, https://doi.org/10.1038/sdata.2017.136, 2017.

Nabil, M., Zhang, M., Bofana, J., Wu, B., Stein, A., Dong, T., Zeng, H., and Shang, J.: Assessing factors impacting the spatial discrepancy of remote sensing based cropland products: a case study in Africa, Int. J. Appl. Earth Obs. Geoinf., 85, 102010, https://doi.org/10.1016/j.jag.2019.102010, 2020.

Waldner, F., Schucknecht, A., Lesiv, M., Gallego, J., See, L., Pérez-Hoyos, A., d'Andrimont, R., De Maet, T., Bayas, J. C. L., Fritz, S., Leo, O., Kerdiles, H., Díez, M., Van Tricht, K., Gilliams, S., Shelestov, A., Lavreniuk, M., Simões, M., Ferraz, R., Bellón, B., Bégué, A., Hazeu, G., Stonacek, V., Kolomaznik, J., Misurec, J., Verón, S. R., De Abelleyra, D., Plotnikov, D., Mingyong, L., Singha, M., Patil, P., Zhang, M., and Defourny, P.: Conflation of expert and crowd reference data to validate global binary thematic maps, Remote Sens. Environ., 221, 235–246, https://doi.org/10.1016/j.rse.2018.10.039, 2019.

Q4. Some of the abbreviations are not explained in detail when they first appear (e.g. LGRIP, CCDC) and it is suggested that the full names be added.

A4: Thank you for pointing this out. We have added the full names of the abbreviations, such as LGRIP and CCDC, upon their first appearance in the manuscript, to improve clarity for readers.

Q5. The terms "cropland" and "farmland" are used interchangeably in the text, and it is suggested that they be standardised as "cropland".

A5: Thank you for pointing this out. We agree that consistent terminology improves clarity. Accordingly, we have standardized the terminology throughout the manuscript and now use the term "cropland" exclusively instead of using it interchangeably with "farmland".

Minor concerns:

Q1. Title: "recent decades of the 21st century" Vague (suggest clarification of year range)

A1: Thank you for your suggestion. In response to the comment and in line with Referee #1's recommendation, we have revised the title to "*An Annual Cropland Extent Dataset for Africa at 30m Spatial Resolution from 2000 to 2022*" to clearly indicate the temporal coverage of the AFCD product.

Q2. The abbreviation "SDG" should be defined at its first occurrence in line 43, rather than being introduced later in line 54.

A2: Thank you for your suggestion. We have added the full term "*Sustainable Development Goals (SDGs)*" at its first occurrence in line 43.

Q3. In Line 25, redundant "for Africa" (appears twice).

A3: Thank you for your suggestion. We have removed the redundant phrase "for Africa" in line 25 to improve clarity.

We have updated the text in Line 26:

"*The study developed a 30-meter resolution African annual cropland distribution (namely AFCD) dataset spanning the years 2000 to 2022.*"

Q4. In Line 41, "croplands play is of critical importance" → "croplands plays a critical role"

A4: Thank you for your suggestion. We have corrected the sentence in line 42 to: "*croplands are of critical importance to global food sustainable development ...*"

Q5. In Line 55, "one in five people undernourished" → "one in five people was undernourished"

A5: Thank you for your comment. We appreciate your suggestion. However, we believe the original phrase "with one in five people undernourished" is grammatically correct and commonly used in scientific writing to express a current condition. Therefore, we have retained the original phrasing in the revised manuscript.

Q6. In Line 76, "GCEP" is undefined and potentially mis-cited (Xiong et al., 2017b refers to GFSAD, not GCEP).

A6: Thank you for pointing this out. We acknowledge the miscitation and have corrected it. The revised sentence now reads:

"*While specialized cropland products, such as the Landsat Global Cropland Extent (Potapov et al., 2022), GFSAD Landsat-Derived Global Rainfed and Irrigated-Cropland Product (Teluguntla et al., 2023), GFSAD Global Cropland Extent Product (Thenkabail et al., 2021), and Digital Earth*"

*Africa (Burton et al., 2022), offer high spatial resolution (ranging from 10 to 30 meters), allowing for detailed landscape characterization.*"

Q7. In Line 97, duplicate use numeric (2).

A7: Thank you for your observation. We have corrected the redundant numbering in Line 97 to ensure consistency and clarity in the text.

Q8. In Line 136, "samples were randomly selected for further validation by students and experts" is vague.

A8: Thank you for your suggestion. In response, we have revised this part of the manuscript to provide a clearer explanation of the validation process, including how the samples were selected and assessed by trained students and domain experts, as also suggested by Referee #1.

Q9. In Line 142, "combines" should be "combined".

A9: Thank you for your suggestion. We have corrected "combines" to "combined" in Line 153 accordingly.

Q10. In Line 177, the third-level heading "3.3.1" should be corrected to "3.2.1"; revise Line 199 to "3.2.2" and Line 223 to "3.3" for consistency.

A10: Thank you for your suggestion. We have corrected the section numbering as follows: "3.3.1" to "3.2.1" in Line 221, "3.2.2" in Line 245, and "3.3" in Line 274 to ensure consistency throughout the manuscript.

Q11. In Line 208, delete "first".

A11: Thank you for your suggestion. We have deleted the word "first" in Line 208 to improve clarity and avoid redundancy in the sentence structure.

Q12. In Line 210 (Equation 1), there appears to be an extraneous "&" symbol that may be a typesetting error. Please verify the mathematical expression's integrity and ensure the formula complies with standard notation conventions.

A12: Thank you for your careful review. We have corrected the typesetting error in Equation (1) by removing the extraneous "&" symbol to ensure the formula adheres to standard mathematical notation.

Q13. In Line 252-254, "The result of this study is developing a new annual cropland dynamic map of Africa at 30-m (Fig. 5). The visual evaluation of the current cropland product shows that cultivated areas are accurately represented across diverse agricultural landscapes throughout Africa (Fig. 5). As shown in the figure above, …" The redundant repeated descriptions of Figure 5 in the text may disrupt the flow; consider consolidating them to strengthen the paragraph's logical coherence.

A13: Thank you for your valuable suggestion. We have carefully revised the paragraph to eliminate redundant references to Figure 5 and improve the logical flow of the text. The updated version reads:

"*The outcome of this study is an annual 30-m cropland dynamics map for Africa (Fig. 5), which demonstrates strong performance in capturing cultivated areas across diverse agricultural landscapes. As illustrated in Figure 5, five representative regions were selected to evaluate the AFCD's capability in recognizing varying cropland patterns under different agricultural conditions.*"

Q14. In Line 304 (Figure 6), The plot elements of the red and blue dots and the purple line should be explained.

A14: Thank you for your suggestion. We have revised the caption of Figure 6 to provide clearer explanations of the plot elements. The updated caption reads:

"*Comparison of AFCD with FAO Statistical Cropland Area (a) and Other LULC/Cropland Products (b-f) (2000–2022). In (b–f), blue and red dots show the differences from FAO estimates*

*for AFCD and other products, respectively. The purple line indicates the linear fit between AFCD and each product.*"

Q15. In Line 415, incorrect serial number.

A15: Thank you for pointing this out. We have corrected the serial number error in Line 415 to ensure consistency throughout the manuscript.